# Training Conditional GANs on Limited and Long-Tailed Data: a Survey and Comparative Analysis

## Abstract

Generative adversarial networks (GANs) are a generative model framework that is competitive with state-of-the-art autoencoders and diffusion models in many tasks. While the latter have achieved impressive generation capabilities, mostly through large-scale, general-purpose text-to-image models, their computational requirements place them out of reach for practitioners. On the other hand, as GAN architectures mature and new developments allow for more stable training, interest in their application has grown across diverse domains. However, real-world data are often hard to deal with due to limited amount of samples or long-tailed distributions. Furthermore, previous works addressing these issues lack guidance regarding their applicability and have not been compared through appropriately diverse benchmarks nor assessed using the same metrics. In this article, we conduct a survey on methods for training GANs on limited and long-tailed data and conduct an extensive comparative analysis of existing methods. Our results allow us to draw conclusions about the advantages, disadvantages, and practical applicability of these methods, hopefully making GANs more accessible to practitioners in diverse fields. The code will be made available as soon as deanonymization is allowed.

## 1 Introduction

Generative Artificial Intelligence, or generative AI, is a particular discipline in deep learning (DL) whose objective is to create novel data by designing models to learn complex patterns and associations from the underlying input distribution. The success of generative AI spans several modalities, such as text, imaging, video, and audio (Lu et al., 2024; DeepSeek-AI et al., 2025; Grattafiori et al., 2024; Huang et al., 2022; Guo et al., 2023; Copet et al., 2023b;a;c; Ramesh et al., 2021; Yan et al., 2021). In this work, we focus on image generation, which aggregates imaging tasks that aim at generating novel content conditioned on some initial image data distribution.

Image synthesis is one of the most representative deep learning tasks, with applications in image translation (Parmar et al., 2023; Tumanyan et al., 2023; Lee et al., 2019), image restoration (Kim et al., 2022; Zhu et al., 2023b; Tran et al., 2020), and image inpainting (Yu et al., 2019; Lugmayr et al., 2022; Suvorov et al., 2022). While much attention is drawn to general-purpose, large-scale models, these require massive computing infrastructure that is not available to most deep learning practitioners. Furthermore, they are trained on datasets containing on the order of millions to billions of images (Deng et al., 2009; Sharma et al., 2018; Schuhmann et al., 2022; Gadre et al., 2023; Zhu et al., 2023a). Datasets of this scale cannot be obtained for many applications due to privacy issues, high data collection costs, and difficulty in homogenizing data from different sources. For example, medical image data can vary significantly between sources depending on the model of the machine used to collect samples, how the machine itself is configured, and how the patient is positioned in the machine. Other issues such as deanonymization, difficulty in sharing data between healthcare institutions, high annotation cost, and naturally different frequencies of anomalies in image data make it very difficult to train generative models for medical image synthesis (Cios & William Moore, 2002). Other domains with similar issues in data collection include brain, satellite, and cellular imaging (Amini et al., 2023). Thus, there is a pressing need for studying and developing new architectures and procedures to train models in limited and imbalanced data.

In this paper, we further restrict our attention to *conditional* deep generative models. While a generative model typically learns the data density $p(\boldsymbol{x})$, conditional generative models learn $p(\boldsymbol{x}|\boldsymbol{y})$, that is, the data distribution conditioned on some information $\boldsymbol{y}$. This additional information can be a class label, a set of attributes that describe the samples, or text. Conditional generative models are particularly useful because they allow for the generation of samples that display specific characteristics or belong to some mode of the data, while typical generative models sample from arbitrary points from the learned data manifold.

There are 3 main state-of-the-art model families that are currently used for image synthesis, conditional or otherwise: variational autoencoders (VAEs), diffusion models, and generative adversarial networks (GANs):

1. VAEs have recently received increased attention due to the capabilities of Vector Quantized VAEs (VQVAEs) and its variations in generating high-quality images (Van Den Oord et al., 2017). This new approach applies a quantized procedure to learn a discrete latent space. Therefore, the posterior distribution is mapped to a discrete distribution. Additionally, VQVAEs solve several problems in VAEs with continuous latent spaces, such as high variance during the generation process and *posterior collapse*(Van Den Oord et al., 2017).

2. Diffusion models define a Markov chain to learn how to reverse a diffusion process, aiming to produce new samples with a distribution similar to that of the training data (Ho et al., 2020). Here, the parametrized chain learns, via variational inference, how to gradually remove noise at each transition until the noise completely vanishes.

3. Generative adversarial networks (GANs) encompass two main models, a generative (G) and a discriminative model (D), denoted as generator and discriminator, respectively (Goodfellow et al., 2014). Both models compete in an adversarial game in which the generator tries to create new samples following the input distribution, while the discriminator attempts to distinguish whether the received image was obtained from the dataset or given by the generator. Under certain conditions, training G and D concurrently forces G to learn how to generate samples with the same quality as the original dataset.

Of the three, diffusion models clearly require larger amounts of data and are more expensive to train (Wang et al., 2023a), making them less viable for many who wish to apply deep generative learning to a particular domain. Indeed, even recent compute-efficient one-step diffusion models struggle to achieve the same quality as GANs even with a larger amount of parameters (Kong et al., 2024; Sordo et al., 2025). Furthermore, while there is an increasing focus on incorporating a GAN-like feedback mechanism into diffusion models to reduce their mode-covering behavior (Karras et al., 2024; Zheng et al., 2025; Yin et al., 2024; Zhou et al., 2024), these still require a comparatively large amount of parameters. Nevertheless, diffusion models remain an exciting research frontier due to their mode coverage and high-quality generation capabilities. We refer the reader to the seminal survey by Croitoru et al. (2023) for an introduction to modern diffusion models, and to Chen et al. (2024) for an up-to-date exposure to the theory behind current conditional diffusion models.

VAEs are regarded as the most stable to train and encourage data coverage due to their objective implicitly minimizing the KL divergence (Shannon, 2020). However, they also tend to generate blurrier samples, and are less popular in limited data regimes (Kebaili et al., 2023; Sordo et al., 2025). GANs typically minimize a *softened reverse KL divergence* (Shannon, 2020) (or a variation thereof), which is more mode-seeking than the KL divergence (Shannon et al., 2020; Li & Farnia, 2023). This behavior explains why GAN samples have such high fidelity and sample quality, though possibly at the cost of ignoring some modes and diversity of the data. For an excellent taxonomy and benchmark of modern GAN architectures and training frameworks, we refer the reader to the seminal work by Kang et al. (2023).

In this work, we focus on (conditional) GANs, which have historically been considered more difficult to train due to their adversarial training regime. Though GANs are poised as a very appealing choice for generative modeling due to their high fidelity, low inference cost, and greater success with limited data, there is currently no unified study of existing methods for training conditional GANs on limited and long-tailed data. Years of development in loss functions and architectures has resulted in GANs that are stable enough that there is now a significant portion of literature dedicated to how to train them in poor data regimes (Zhao et al.,

2020b; Tran et al., 2021; Zhao et al., 2020a; Karras et al., 2020a; Jiang et al., 2021; Tseng et al., 2021; Shahbazi et al., 2022; Rangwani et al., 2022; Khorram et al., 2024; Rangwani et al., 2023). Furthermore, though VAEs typically achieve greater data coverage, they also tend to result in lesser fidelity to the real data, generating blurry and out-of-distribution samples more often than GANs.

Our objective in this work is to analyse state-of-the-art methods to train conditional GANs in limited and long-tailed data, which are a common setting in real-world applications (Zhang et al., 2023). By long-tailed data, we refer to imbalanced datasets in which a significant fraction of samples belong to a few *head* classes, while a vast majority of *tail* classes have a relatively much smaller amount of samples. We note a lack of guidance in previous works as to when these methods should be applied, and argue that they have not yet been compared through appropriately diverse benchmarks nor assessed consistently through the same metrics. These issues make the advantages and disadvantages of each model, as well as their applicability, unclear to practitioners. As such, we devise a rigorous evaluation scheme that we use to compare the models and also formalize the advantages and disadvantages of each.

With our proposed systematic evaluation, we observe that limited data causes training to be unstable, and that feature sharing methods, which learn global features that are shared across several classes, are particularly effective at ameliorating such issue. In long-tailed data, feature sharing methods also demonstrate improved generative capabilities for the tail of the data distribution.

The paper is organized as follows. We start by giving an overview on GANs and how they are trained conditionally, as well as describing how limited and long-tailed data affect their training process, in Section 2. In Section 3 we provide a literature review of existing methods for addressing the issues of training conditional GANs in limited and long-tailed data. These methods are assessed according to a set of diverse datasets and metrics, which are described in detail in Section 4. In Section 5 we present the results of this evaluation and discuss them in the context of each method, arriving at a clearer understanding of how/when they work, their advantages and disadvantages, and thus their applicability. Finally, in Section 6, we present some practical considerations for those wishing to apply these methods to their domain, as well as suggestions for future work based on our findings.

Our contributions are as follows:

1. An extensive literature review of existing methods for training conditional GANs in limited and long-tailed data, categorized by how they alter the training process: through data augmentation, knowledge sharing, and explicit regularization.

2. The first extensive comparative analysis of these methods in a diverse set of datasets, allowing for a greater understanding of their practical applicability. We evaluate these methods using standard, widely accepted metrics that not only consider the learned models' overall quality, fidelity, and diversity, but also whether these qualities are observed in classes/modes with little representation (i.e., tail classes).

3. Based on the empirical results from these benchmarks, we present a discussion on the practical applicability of these methods. Furthermore, we provide recommendations for practitioners and directions for future research.

4. We provide a codebase containing implementations of all the methods and metrics described, including instructions for generating the datasets, in order to make reproducibility as easy as possible.

We hope that these contributions will make these methods more accessible to both machine learning researchers developing new methods and to practitioners from other fields who might benefit from the generative abilities of conditional GANs in their own domains.

## 2 Background

This section provides an overview of fundamental concepts relevant to the rest of the paper. Section 2.1 introduces the mathematical framework behind generative adversarial networks (GANs), and Section 2.2

shows how GANs can be extended for conditional training. The problems of training conditional GANs on limited and long-tailed data are presented in Sections 2.3 and 2.4, respectively.

## 2.1 Generative Adversarial Networks

The GAN framework (Goodfellow et al., 2014) consists of simultaneously training two neural networks in an adversarial game. The *generator* network $G$ and the *discriminator* network $D$ are optimized jointly in the following manner: given some vector of random noise as input, $G$ generates a fake example. $D$, in turn, must learn to distinguish between real and fake examples, outputting a prediction (e.g., a probability) of how real a given example looks. This provides feedback to $G$ so that it learns to generate examples that are closer to the distribution of the real data. We can formalize this adversarial training regime as follows:

$$\min_G \max_D \mathbb{E}_{\boldsymbol{x} \sim p_{\text{data}}} [f(D(\boldsymbol{x}))] + \mathbb{E}_{\boldsymbol{z} \sim p_{\boldsymbol{z}}} [f(-D(G(\boldsymbol{z})))] . \tag{1}$$

In equation 1, $\boldsymbol{x} \sim p_{\text{data}}$ is a real example from the training set and $\boldsymbol{z}$ is a random vector sampled from a tractable distribution $p_{\boldsymbol{z}}$, usually the standard Gaussian, such that $\boldsymbol{z} \sim \mathcal{N}(\boldsymbol{0}, \boldsymbol{1})$. $G(\boldsymbol{z})$ is then a fake example generated from a noise vector $\boldsymbol{z}$, and $D(\cdot)$ is some score of realness given by the discriminator. Finally, $f(\cdot)$ is simply some transformation of the outputs of the discriminator that depends on the loss function being used (Liu et al., 2020); if $f(x) = -\log(1 + e^{-x})$, we recover the original GAN loss (Goodfellow et al., 2014). We can write the loss for the discriminator and generator, respectively $\mathcal{L}^D$ and $\mathcal{L}^G$, as follows:

$$\mathcal{L}^D = -\mathbb{E}_{\boldsymbol{x} \sim p_{\text{data}}} [f(D(\boldsymbol{x}))] - \mathbb{E}_{\boldsymbol{z} \sim p_{\boldsymbol{z}}} [f(-D(G(\boldsymbol{z})))]$$
$$\mathcal{L}^G = \mathbb{E}_{\boldsymbol{z} \sim p_{\boldsymbol{z}}} [f(-D(G(\boldsymbol{z})))]$$

## 2.2 Conditional GANs

Conditional GANs (CGANs) extend the GAN framework by incorporating additional information $\boldsymbol{y}$ (e.g., class labels or sample attributes) into the training process. That is, instead of learning the data density $p(\boldsymbol{x})$ we learn $p(\boldsymbol{x}|\boldsymbol{y})$. For simplicity, moving forward we assume $\boldsymbol{y}$ are class labels, but all that follows can be generalized for other conditioning signals such as image attributes (White, 2016) or text (Carlini et al., 2023b).

By learning this conditional data distribution, we can draw samples conditioned on some $\boldsymbol{y}$, e.g., generating images of animals that belong to a certain species. To that end, we modify the losses for the discriminator and generator to make them conditional on $\boldsymbol{y}$:

$$\mathcal{L}_c^D = -\mathbb{E}_{\boldsymbol{x} \sim p_{\text{data}}} [f(D(\boldsymbol{x}))] + \mathbb{E}_{\boldsymbol{z} \sim p_{\boldsymbol{z}}} [f(D(G(\boldsymbol{z}, \boldsymbol{y})))] \tag{2}$$
$$\mathcal{L}_c^G = \mathbb{E}_{\boldsymbol{z} \sim p_{\boldsymbol{z}}} [f(D(G(\boldsymbol{z}, \boldsymbol{y})))] , \tag{3}$$

where $\boldsymbol{y}$ is usually sampled according to its frequency in the training data (instance-balanced sampling).

The way $\boldsymbol{y}$ is injected into the networks differs depending on the base model. Conditional batch normalization (CBN) (Dumoulin et al., 2017; de Vries et al., 2017) modifies batch normalization layers in the generator so that the gain and bias depend on additional information. When the additional information $\boldsymbol{y}$ corresponds to class labels, this is done by having a separate set of batch normalization parameters for each class. In other words, CBN induces condition-specific features at each layer. Models using CBN include (Brock et al., 2019; Miyato & Koyama, 2018; Miyato et al., 2018; Zhang et al., 2019a; Kang & Park, 2021; Kang et al., 2021; Hou et al., 2022).

Another possibility to inject conditional information is through style vectors. Traditional GAN generators are a function from the latent space $\mathcal{Z}$ directly to the data space $\mathcal{X}$, i.e., $G : \mathcal{Z} \to \mathcal{X}$. StyleGAN architectures (Karras et al., 2019; 2020b) have a *mapping network* $M : \mathcal{Z} \to \mathcal{W}$, which takes as input the usual latent noise $\boldsymbol{z}$ and maps it to so-called *style vectors* $\boldsymbol{w}$ in an intermediate space $\mathcal{W}$, and a *synthesis network* $S : \mathcal{W} \to \mathcal{X}$, which maps the style vectors $\boldsymbol{w}$ to images. Thus, the StyleGAN generator is $G = S \circ M$. The mapping network is fully connected and, in practice, the learned intermediate representation $\mathcal{W}$ is found

to be semantically meaningful and disentangled (Shen et al., 2020; Collins et al., 2020). The synthesis network is a transpose convolutional network, where style vectors $\boldsymbol{w}$ are introduced at each convolutional layer via adaptive instance normalization (Huang & Belongie, 2017), allowing for control over the output at all resolutions. In conditional settings, a condition vector $\boldsymbol{y}$ will usually be concatenated to the noise vector $\boldsymbol{z}$ and passed through the mapping network, i.e., $M : \mathcal{Z} \times \mathcal{Y} \to \mathcal{W}$. The resulting style vectors $\boldsymbol{w}$ encode the conditional information and influence the image synthesis process at each step as described above.

Conditional batch normalization and style vectors are the two most successful ways to inject conditional information into the *generator*. We now examine the two prevalent methods for injecting conditional information into the discriminator.

Classifier-based CGANs (Odena et al., 2017; Mariani et al., 2018; Gong et al., 2019; Kang & Park, 2021; Hou et al., 2022; Kang et al., 2021) introduce class discrimination explicitly into the objective. The earliest and simplest of these methods is ACGAN (Odena et al., 2017), which adds a classifier head to the discriminator and computes the usual classification cross-entropy loss, which is then added to the GAN objective. With a little abuse of notation, with $D(\cdot)$ now representing a class prediction instead of a classification between real and fake, the loss of a classifier-based CGAN discriminator becomes:

$$\mathcal{L}_c^D = \mathcal{L}^D + \mathrm{CE}\big(D(\boldsymbol{x}), \boldsymbol{y}\big) + \mathrm{CE}\big(D(G(\boldsymbol{z}, \boldsymbol{y})), \boldsymbol{y}\big), \tag{4}$$

where CE denotes the categorical cross-entropy.

However, most GANs use a *projection discriminator* (Miyato & Koyama, 2018) in order to infuse class information into the discriminator. The projection discriminator takes the conditional information $\boldsymbol{y}$ and maps it to an embedding $\boldsymbol{e}_y$. Then, it computes the dot product between the features of the penultimate layer of the discriminator (i.e., before the classification layer) and the embedded conditional information $\boldsymbol{e}_y$, which yields the projection discriminator loss. It is shown in (Miyato & Koyama, 2018) that this form of the discriminator arises from the assumption that $p(\boldsymbol{y}|\boldsymbol{x})$ is unimodal, which the authors argue is common in class-conditional image generation. Projection discriminators are used in most representative GAN architectures (Miyato et al., 2018; Brock et al., 2019; Karras et al., 2019; 2020b).

## 2.3 The Limited Data Problem

Part of the success of GANs hinges on its large amount of parameters, which allows them to approximate very complex data densities. However, as GANs see wider adoption and applicability, some issues arise when dealing with limited data, since the discriminator has enough capacity to memorize the training samples. In this scenario, the discriminator can easily classify a sample as being fake if it is trivially dissimilar to the samples in the training data. This results in a degenerate solution where in order to optimize its objective to fool the discriminator, the generator converges to a point where the produced images are extremely similar to the training data. This failure mode is known as *mode collapse*. In this scenario, GANs generate samples with very little diversity, and may omit modes of variability (e.g., classes) altogether. In the worst-case scenario, the generator is completely unable to improve from the feedback of the overfitted discriminator, and training diverges.

Since earlier GAN developments were dedicated to the general unconditional training scenario, the term *mode collapse* has historically carried a different meaning. In practice, the learned generator manifold and the data manifold are expected to be disjoint (Arjovsky & Bottou, 2017). Since $D$ tends to be close to 1 near most real samples (Denton et al., 2015), the more populated modes will have a higher probability of attracting the gradient of the generator. Furthermore, once the (unconditional) generator finds some subset of the data modes, it is not penalized for the ones it is missing (Che et al., 2016). This results in unconditional GANs identifying only some modes of variability the discriminator believes highly likely and placing all their probability mass there, silently ignoring the others (Metz et al., 2017). This failure mode, where GANs are unable to recover some of the modes of the real data, is currently known as *mode dropping* (Yazici et al., 2020).

In conditional training, GANs are explicitly encouraged to visit all the modes of variability present in the data, e.g., when training a class-conditional GAN, class information is a very strong prior which usually

guarantees that the generator visits each class. However, when data for a given class are limited, the discriminator is usually powerful enough to quickly memorize them, so that the only way for the generator to not be penalized is to generate near-exact fakes. We reiterate that this failure mode, where for a given class the GAN outputs samples with very little diversity (but possibly high quality) is known as *mode collapse*. Though mode collapse happens at a large scale when the data is limited, data imbalances can make it so that mode collapse is particularly difficult to deal with in specific classes, as we discuss next.

### 2.4 The Long-Tailed Data Problem

It is commonly the case that real data follow a so-called *long-tailed* distribution, whereby a large portion of the examples correspond to only a few classes, while most classes have very few representative samples (Cui et al., 2019; Zhang et al., 2023). The former are called *head classes*, and the latter are called *tail classes*. Models, both generative and discriminative, trained on long-tailed data tend to become biased towards the head classes and perform very poorly on tail classes. We note that our use of the term *long-tailed* follows established usage in the machine learning literature, and differs from the statistical notion of heavy-tailed distributions. We use it to refer to datasets where most samples are concentrated in a few head classes, while the remaining classes have comparatively few samples.

It had previously been observed that CGANs had a tendency to become invariant to the noise inputs, i.e., producing similar outputs for different $z$, which was generally attributed to class conditioning being too strong of a prior (Liu et al., 2019; Firman et al., 2018; Mao et al., 2019; Yang et al., 2019). Based on the observation that conditional generative image modeling is ill-defined (i.e., there is more than one correct output for each condition), these methods enforce the generator's outputs to become sensitive to changes in the latent space, leading to more diverse outputs in a class-conditional setting.

Particularly for CGANs trained on long-tailed data, we observe that the discriminator tends to memorize the few examples in the tail classes, giving meaningless feedback to the generator. This leads to the afore-mentioned "mode collapse" scenario, where in order to fool the discriminator, the generator learns to simply reproduce the training examples of the tail classes with very little variability.

Finally, recent works observed that for a set of semantically similar classes, if one of them has significantly greater representation, *class confusion* may occur, where a CGAN will generate samples belonging to the majority classes within this set when tasked with generating samples from the minority ones (Rangwani et al., 2022; 2023).

## 3 Literature Review

In this section, we extensively review methods that address the problems of training CGANs on limited and long-tailed data, as described, respectively, in Sections 2.3 and 2.4. We divide these methods into three categories — data augmentation, knowledge sharing, and regularization — according to how they affect the learning process.

### 3.1 Data Augmentation

In deep learning classification tasks, the usual solution for limited or imbalanced data is to apply data augmentations. This is done by applying transformations that make the data harder to classify while preserving their semantics. The idea is that the classifier will become invariant to these transformations, resulting in an increased generalization ability. However, in generative modeling tasks, this method presents an issue: that the model might learn the augmented distribution.

Data augmentation methods for GANs hinge on the concept of consistency regularization. First popularized in the semi-supervised learning literature (Fan et al., 2023), consistency regularization enforces that a model should yield the same outputs for augmented versions of one ground truth. This is done by introducing an auxiliary objective into the training process such that the model becomes invariant to these augmentations.

Initial applications of consistency regularization to GANs acted only on the discriminator, making it invariant to semantics-preserving transformations applied to real images (Wei et al., 2018; Zhang et al., 2019b). An issue with these methods is that since the discriminator yields similar outputs for both real and augmented real images, the generator might include augmentation artifacts in fake images to fool the discriminator since it might associate those artifacts with real images. This was shown to happen in practice by Zhao et al. (2021), who propose *balanced consistency regularization* (bCR). This extends previous methods by also making the discriminator invariant to transformations of generated images, so that the generator no longer has an incentive to include augmentation artifacts in the generated images.

It is shown experimentally that even though bCR does not encourage the generator to include augmentations in the fake images, there is no penalty for doing so, and thus augmentations still appear spuriously in the generated images (Karras et al., 2020a). The authors introduce *adaptive discriminator augmentation* (ADA), which instead of regularizing the discriminator to be invariant to augmentations, only trains it on augmented images, both real and fake.

The idea behind ADA is that GANs are still able to find the correct distribution as long as the applied augmentations are *non-leaking*. By applying a given augmentation with probability $p < 1$, the relative frequency of examples without that augmentation is greater, and thus the model should in theory be able to learn the underlying non-augmented distribution. They show that several data-space augmentations rarely ever leak as long as they are executed with $p < 0.8$. This value is initialized to 0 and set adaptively during training depending on the performance of the discriminator in classifying real (augmented) images. It is the probability with which to apply each augmentation from a predefined set to a given (real or fake) image. Note that even for small values of $p$, given a large enough set of augmentations, the discriminator will rarely see a non-augmented image.

Other discriminator augmentation methods (Zhao et al., 2020b; Tran et al., 2021; Zhao et al., 2020a) were developed concurrently with ADA, but these do not set augmentation strength adaptively and seem to, at best, achieve comparable performance (Karras et al., 2020a; Wang et al., 2023b; Carlini et al., 2023a; Kang et al., 2023). Nevertheless, out of the three concurrent works (Zhao et al., 2020b; Tran et al., 2021; Zhao et al., 2020a), DiffAug (Zhao et al., 2020a), a differentiable data augmentation method, has seen the most adoption (Sauer et al., 2021; Tseng et al., 2021; Carlini et al., 2023b).

Negative data augmentation (Sinha et al., 2021; Zhang et al., 2024) uses prior knowledge about out-of-distribution samples to guide the generator's support to not over-generalize. In particular, it applies non-semantics-preserving image augmentations to real images, increasing the precision (Sajjadi et al., 2018; Kynkäänniemi et al., 2019) of the discriminator. This form of weak supervision reduces the amount of *false positives* by explicitly guiding the generator's support away from regions that should have zero probability. While these methods are effective in preventing the generator from placing probability mass in regions it is not supposed to, this is generally not a problem that ails GANs trained with limited or long-tailed data, where the issue has more to do with coverage of all the modes of variability in the data.

As a final note, the methods covered in this section were based on consistency regularization, which force the discriminator to be invariant to semantics-preserving transformations of the data. Despite the *regularization* in the name, we distinguish these methods from the ones in Section 3.3 since while these all apply transformations in the data space (Wei et al., 2018; Zhang et al., 2019b; Zhao et al., 2021; 2020b; Tran et al., 2021; Zhao et al., 2020a; Karras et al., 2020a; Sinha et al., 2021; Zhang et al., 2024), the ones presented later consist of alterations to the training objective so as to explicitly regularize the model.

### 3.2 Knowledge Sharing

Knowledge sharing methods for training conditional GANs change the training procedure so that the model's ability to learn certain modalities of the data benefits from the diversity found in the entire dataset.

When training conditional GANs, we might expect the additional (e.g., class) supervision to improve training. However, as first observed by Shahbazi et al. (2022), training conditional GANs on small datasets is more difficult than their unconditional counterparts. The authors show that in datasets where unconditional GANs would have satisfactory performance, conditional training induces severe mode collapse. In practice,

conditional training seems to constrain GANs to distribute probability mass more conservatively around each class, resulting in reduced diversity.

Transitional-CGANs (Shahbazi et al., 2022) start by training the GAN in a completely unconditional manner, taking advantage of more stable training and greater diversity, and eventually gradually start transitioning to conditional training. Conditional training is done through an auxiliary classification objective, where the discriminator tries to predict the class of a given (real or fake) sample (Odena et al., 2017). The model has two training objectives: the usual discriminative GAN objective (Eq. 1) with loss $\mathcal{L}_{\mathrm{uc}}$ and a classification objective as in AC-GAN (Odena et al., 2017), with loss $\mathcal{L}_{\mathrm{c}}$ (cf. the second and third terms of Eq. 4), which result in what the authors dub "unconditional and conditional losses", respectively $\mathcal{L}_{\mathrm{uc}}$ and $\mathcal{L}_{\mathrm{c}}$. The Transitional-CGAN total loss is then:

$$\mathcal{L} = \mathcal{L}_{\mathrm{uc}} + \lambda_t \mathcal{L}_{\mathrm{c}}, \tag{5}$$

where $\lambda_t \in \mathbb{R}$ controls the strength of the conditional training. This value is not itself a hyperparameter, but rather depends on a specified start and end of the transition to conditional training, defined in terms of a number of training iterations, respectively $T_s$ and $T_e$. Note that the unconditional objective is retained throughout training; the transition is only with respect the the conditional loss. At any iteration $t$ during training, we have:

$$\lambda_t = \min\left(\max\left(\frac{t - T_s}{T_e - T_s}, 0\right), 1\right). \tag{6}$$

Importantly, the value of $\lambda$ will be 0 until the transition starts, meaning that training is completely unconditional at that point. During the transition, $\lambda_t$ increases linearly until it reaches its maximum value of 1, where it remains for the rest of the training.

We note that while usually the input to StyleGAN's mapping network is a concatenation of the random noise vector $\boldsymbol{z}$ and the conditional information vector $\boldsymbol{y}$, Transitional-CGAN requires that they are instead added together in order to make the transition work, i.e., the input to the mapping network in Transitional-CGAN is $\boldsymbol{z} + \boldsymbol{y}$. In isolation, this change worsens model performance. Ultimately, whether Transitional-CGAN is useful depends on whether its feature sharing properties outweigh this performance degradation.

Requiring only the specification of the start and end of the transition, Transitional-CGAN consistently improved generation quality and diversity in datasets with at most a few thousand total images. All datasets were either perfectly or reasonably balanced, so it is unclear how this method would extend to imbalanced or, in the limit, long-tailed datasets.

More recently, Khorram et al. (2024) propose unconditional training at lower resolutions (UTLO) to address the challenge of training GANs on long-tailed data. They train the lowest-resolution layers in both the generator and discriminator in a completely unconditional manner, based on the observation that similarities between head and tail classes are often found at lower resolutions. The goal is that the generation of images from tail classes can benefit from the diversity found in head classes.

The generation process is modified as follows: for each latent vector $\boldsymbol{z}$, two style vectors are generated. One of them is computed as usual, by concatenating $\boldsymbol{z}$ and conditional information $\boldsymbol{y}$ and passing them through the mapping network, getting a vector $\boldsymbol{w}_y$ conditioned on $\boldsymbol{y}$. For the other style vector, $\boldsymbol{y}$ is set to zero, resulting in a vector $\boldsymbol{w}$ that is not influenced by any conditional information.

The unconditional style vector is injected into the first few layers of the synthesis network, where at a specified resolution we collect the resulting intermediate unconditional image. [1] After that point, the forward pass through the generator resumes as usual, starting from the intermediate unconditional feature maps and now using style vectors with conditional information.

In the discriminator, the unconditional intermediate resolution images are inserted at the layer with corresponding resolution, resulting in outputs that are not affected by conditional information.

Note that in the generator, conditional generation starts from the intermediate output of the unconditional stump, so that the low- and full-resolution images are generated in a single forward pass. In the discriminator,

---

[1] Many generator architectures, including that of StyleGAN2, use skip connections where an RGB image is generated at each intermediate resolution.

two distinct forward passes occur: a full pass for the high-resolution fake image, and one that starts only at the chosen intermediate resolution of the unconditional image.

Denoting the loss of the above unconditional objective $\mathcal{L}_{\text{uc}}$ and the usual conditional (e.g., projection discriminator (Miyato & Koyama, 2018)) loss $\mathcal{L}_{\text{c}}$, we can write the UTLO loss:

$$\mathcal{L} = \mathcal{L}_{\text{c}} + \lambda \mathcal{L}_{\text{uc}}. \tag{7}$$

UTLO requires specifying the resolution at which we retrieve unconditional images (usually $8 \times 8$) and the weight of the unconditional objective $\lambda \in \mathbb{R}$. Importantly, while UTLO might benefit from knowledge sharing at low resolutions, we cannot exclude the possibility that it introduces out-of-sample high-level features to the generated images, e.g., generating tail class objects in backgrounds or poses that are unnatural. We determine experimentally that this does not seem to happen in practice, and discuss how UTLO impacts model precision in Section 6.

Finally, we note that the loss functions resulting from each of these knowledge sharing methods apply to both the generator and the discriminator.

### 3.3 Regularization

Regularization methods for dealing with limited and long-tailed data modify the training objective, either adaptively, according to the training dynamics, or by modifying the training process, using prior knowledge about the class distribution.

Based on the observation that discriminators tend to get overconfident or memorize training examples for certain modes, it was proposed in Tseng et al. (2021) to encourage the discriminator's outputs for real and fake images to be similar. This method is known as the LeCam regularizer, and gets its name from the fact that under mild assumptions, optimizing WGANs (Arjovsky et al., 2017; Gulrajani et al., 2017) using this regularizer amounts to minimizing a weighted LeCam-divergence.

The LeCam regularizer makes use of two exponential moving averages of the discriminator outputs for real and fake images, $\alpha_R$ and $\alpha_F$, respectively. Due to the instability of discriminator outputs, LeCam regularization is applied only after a warm-up period for these to stabilize. The regularization term is:

$$\mathcal{L}_{\text{LeCam}} = \mathbb{E}_{\boldsymbol{x}}\left[\|D(\boldsymbol{x}) - \alpha_F\|_2^2\right] + \mathbb{E}_{\boldsymbol{z}}\left[\|D(G(\boldsymbol{z})) - \alpha_R\|_2^2\right] \tag{8}$$

While the above may seem counterintuitive, penalizing the discriminator for being overconfident reduces mode collapse because the generator is not so aggressively penalized for generating samples that do not look very similar to the training data. This is particularly useful in imbalanced datasets, where minority classes may benefit from the generator having more freedom to expand its support for those classes. The above loss is added to the discriminator's loss:

$$\mathcal{L}^D \leftarrow \mathcal{L}^D + \lambda \mathcal{L}_{\text{LeCam}}, \tag{9}$$

where $\lambda \in \mathbb{R}$ dictates the strength of the regularizer. Adding LeCam regularization to BigGAN (Brock et al., 2019) resulted in significant improvements in FID; however, when added to StyleGAN2 (Karras et al., 2020b) with adaptive data augmentation (ADA) (Karras et al., 2020a), the improvements were more modest. The choice of $\lambda$ is highly dependent on the base architecture, with values between $[0.1, 0.5]$ being used in BigGAN and values as small as $3 \times 10^{-7}$ being used in StyleGAN2.

Rangwani et al. (2022) noted that while data augmentation methods and other regularization techniques helped mitigate mode collapse in limited data settings, that was not the case for long-tailed data. When grouping the class-specific conditional batch normalization (CBN) parameters into a matrix, they observe that its spectral norm (i.e., its largest singular value) explodes for tail classes.

Writing the CBN gains and biases for layer $l$ and class $y$ as $\boldsymbol{\gamma}_{l,y}, \boldsymbol{\beta}_{l,y} \in \mathbb{R}^{F_l}$, with $F_l$ the number of features in layer $l$, the authors group these parameters into $g$ groups and $c$ columns (such that $g \times c = F_l$). Exemplifying using the gain parameters, this arrangement yields a matrix $\boldsymbol{\Gamma}_{l,y} \in \mathbb{R}^{g \times c}$. They find that the spectral norm $\sigma_{\max}(\boldsymbol{\Gamma}_{l,y}) \in \mathbb{R}$ explodes for values of $y$ corresponding to the tail classes, which is a symptom of the same

features always being induced when the generator is asked to generate a sample belonging to these classes. Given gain and bias parameter groups $\boldsymbol{\Gamma}_{l,y}, \boldsymbol{B}_{l,y} \in \mathbb{R}^{g \times c}$ for layer $l$ and class $y$, the so-called group spectral regularization (GSR) loss is:

$$\mathcal{L}_{\text{GSR}} = \sum_l \sum_y \lambda_y \left( \sigma_{\max}^2(\boldsymbol{\Gamma}_{l,y}) + \sigma_{\max}^2(\boldsymbol{B}_{l,y}) \right), \tag{10}$$

where $\lambda_y \in \mathbb{R}$ is (softly) inversely related to the number of samples $n_y$ of class $y$, i.e., $\lambda_y$ is the effective number of samples (Cui et al., 2019):

$$\lambda_y = \frac{1 - \alpha}{1 - \alpha^{n_y}}, \tag{11}$$

where $\alpha$ is a hyperparameter. This loss is applied to the generator:

$$\mathcal{L}^G \leftarrow \mathcal{L}^G + \lambda_{\text{GSR}} \mathcal{L}_{\text{GSR}}, \tag{12}$$

where the regularizer strength $\lambda_{\text{GSR}}$ was set to 0.5 for all their experiments when added to BigGAN, SNGAN (Miyato et al., 2018), and StyleGAN2. FID (Heusel et al., 2017), a commonly used metric for assessing generative image models, was consistently improved, and the method seems very robust to the choice of $\lambda_{\text{GSR}}$ as well as to the choice of how to group the parameters.

We also note that since StyleGAN2 does not use CBN, the authors recommend applying GSR by constraining the norm of the style vectors $\boldsymbol{w}$. Given a batch of style vectors $\boldsymbol{w} \in \mathbb{R}^{512}$, each vector is grouped to $16 \times 32$ and the resulting spectral norm is penalized. Because the $\boldsymbol{w}$ can amplify certain directions in the synthesis network's feature space, applying GSR in this manner can result in style vectors that have less extreme eigenvalues and whose dimensions are less correlated.

Interestingly, the authors argue that there is a greater need for GSR in the earlier layers of the generator as they encode class-specific information; this is exactly the opposite of the assumption made in UTLO (Khorram et al., 2024), which trains the earlier layers of the generator in a completely unconditional manner, arguing that the global/coarse information they encode is class-generic. Furthermore, visualizations of the covariance between CBN parameters seem to go against the authors' point, as they show larger covariances between grouped parameters in later (i.e., higher resolution) layers of the generator when GSR is not applied (Rangwani et al., 2022).

While it was known that GAN outputs became insensitive to the input noise vectors $\boldsymbol{z}$ in the presence of limited data, the authors of (Rangwani et al., 2023) observed that the vectors in the intermediate latent space of StyleGANs also collapsed in the class-conditional setting, and that although GSR (Rangwani et al., 2022) successfully reduced mode collapse, it resulted in class confusion between semantically similar classes.

The authors hypothesize that one reason that the style vectors become insensitive to changes in $\boldsymbol{z}$ is that while the latter are continuous variables (usually a standard normal), the class labels $\boldsymbol{c}$ are discrete, leading GANs to converge to a degenerate solution of generating the same image for each class. They argue that inducing continuity in the conditioning vectors would be beneficial, and propose augmenting the conditioning vectors with noise inversely proportional to sample frequency:

$$\tilde{\boldsymbol{c}} = \boldsymbol{c} + \mathcal{N}(\boldsymbol{0}, \boldsymbol{\sigma}_c \, \mathbb{I}), \tag{13}$$

where $\boldsymbol{\sigma}_c = \sigma \boldsymbol{\lambda}_y$, $\sigma \in \mathbb{R}$ is a hyperparameter, and $\boldsymbol{\lambda}_y$ are the class-wise effective numbers of samples (Cui et al., 2019) (cf. Eq. 11). The vector $\tilde{\boldsymbol{c}}$ is then concatenated with $\boldsymbol{z}$ as usual. Through this augmentation, the style vectors become more diverse, which results in greater image diversity.

However, while the above expands the support for each class in the intermediate space $\mathcal{W}$, overlaps between these regions could result in class confusion. In order to enforce invariance to the noise augmentation above, and based on the Barlow Twins method for self-supervised learning (Zbontar et al., 2021), the authors introduce another method: for each class vector $\boldsymbol{c}$, two *twin* augmentations $\tilde{\boldsymbol{c}}_a, \tilde{\boldsymbol{c}}_b$ are generated and concatenated to the same latent noise $\boldsymbol{z}$. They are called twin augmentations since they result from adding two different noise vectors to a common one-hot class vector. This results in two batches of *twin* augmented latents $\tilde{\boldsymbol{W}}_a, \tilde{\boldsymbol{W}}_b$, from which we compute a cross-correlation matrix $\boldsymbol{C}$, which is square and whose

dimensionality along each axis is the same as that of the style vectors. The final method is called NoisyTwins, and its loss is:

$$\mathcal{L}_{\mathrm{NT}} = \sum_i (1 - \boldsymbol{C}_{i,i}^2) + \gamma \sum_{i \neq j} \boldsymbol{C}_{i,j}^2, \tag{14}$$

where $\gamma$ is a hyperparameter. The first term encourages the twin style vectors (i.e., each $\boldsymbol{w}_a, \boldsymbol{w}_b$ pair originating from the same class label) to be similar, while the second tries to maximize information in the style vectors by decorrelating its variables. The NoisyTwins loss is added to the generator loss, scaled by another hyperparameter $\lambda \in \mathbb{R}$:

$$\mathcal{L}^G \leftarrow \mathcal{L}^G + \lambda \mathcal{L}_{\mathrm{NT}} \tag{15}$$

## 4 Methodology

We perform extensive experiments assessing the performance of the methods described above, testing them on a diverse set of datasets, using a common architecture as a backbone, and reporting several standard metrics. This allows us to clearly state the advantages and disadvantages of each method, as well as to establish guidelines for practical application.

In this section, we thoroughly describe the experimental setting, including training methods, datasets, and metrics, used to evaluate the methods presented in Section 3.

### 4.1 Models

We use StyleGAN2 as the backbone model for our experiments. It has been found to generally be competitive with BigGAN in large-scale studies (Kang et al., 2023) and has been reported to produce more predictable results (Karras et al., 2020a). Furthermore, it accommodates the NoisyTwins method, which is unique to the StyleGAN architecture. Finally, Shahbazi et al. (2022) report that BigGAN struggles to learn in limited data settings, both conditional and unconditional (Shahbazi et al., 2022).

We add adaptive discriminator augmentation (ADA) (Karras et al., 2020a) to StyleGAN2 to form a strong baseline, as it has been found to consistently outperform other data augmentation methods (Kang et al., 2023). Starting from the official implementation of StyleGAN2+ADA[2], we add Transitional-CGAN[3] (Shahbazi et al., 2022), unconditional training at lower resolutions (UTLO)[4] (Khorram et al., 2024), the LeCam regularizer (Tseng et al., 2021),group spectral regularization (GSR)[5] (Rangwani et al., 2022), and NoisyTwins[6] (Rangwani et al., 2023). Details about hyperparameter choices can be found in the Appendix.

### 4.2 Datasets

We use six datasets to test the above models. They contain varying sample sizes and imbalances, which allows us to study the methods' performances in different data availability circumstances:

- **ImageNet Carnivores** is a subset of the ImageNet (Deng et al., 2009) dataset containing only classes corresponding to carnivore animals. We use the same subset of the Carnivores subset as Shahbazi et al. (2022), which contains 20 classes and 100 images per class, i.e., it is perfectly balanced.

- **AnimalFaces-LT** originally contains 20 classes of animals with a total of $2,409$ images. Similarly to Khorram et al. (2024), we set its *imbalance ratio* to be $\rho = 25$, resulting in a total of $1,756$. The imbalance ratio $\rho$ of a dataset is the ratio of the number of samples between the class with the most samples and the class with the least samples. The *-LT* suffix denotes that it has been modified to increase its degree of "long-tailedness". The intermediate classes are assigned sizes that decrease geometrically from head to tail, resulting in an exponential distribution of samples across classes.

---

[2]https://github.com/NVlabs/stylegan2-ada-pytorch
[3]https://github.com/mshahbazi72/transitional-cGAN
[4]https://github.com/khorrams/utlo
[5]https://github.com/val-iisc/gSRGAN
[6]https://github.com/val-iisc/NoisyTwins

Table 1: Summary of datasets used in our experiments.

| Dataset | N. Samples | N. Classes | Imb. ratio $\rho$ | Resolution | Characteristics |
|---|---|---|---|---|---|
| Carnivores | $2,000$ | 20 | 1.0 | $64 \times 64$ | Limited, balanced |
| AnimalFaces-LT | $1,756$ | 20 | 25.0 | $64 \times 64$ | Limited, long-tailed |
| iNaturalist2019 | $268,243$ | $1,010$ | 31.25 | $64 \times 64$ | Long-tailed, fine-grained |
| Flowers-LT | $5,285$ | 102 | 100 | $128 \times 128$ | Limited, long-tailed |
| ImageNet-LT | $115,846$ | $1,000$ | 100 | $64 \times 64$ | Long-tailed |
| CIFAR10-LT | $12,406$ | 10 | 100 | $32 \times 32$ | Long-tailed |

- **iNaturalist2019** (Horn et al., 2019) is a naturally long-tailed ($\rho = 31.25$) dataset with around $268,243$ images and $1,010$ classes. The dataset was first used in a fine-grained and long-tailed recognition task. There is a large degree of similarity between many of the classes since they often share textures and colors.

- **Flowers-LT** (Nilsback & Zisserman, 2008) contains 102 flower categories and, originally, $8,189$ images. We again increase the imbalance ratio to $\rho = 100$, resulting in a dataset with $5,285$ images. This dataset has the lowest inter-class variability, since all its classes are flowers, which have very similar global structures, backgrounds, and textures.

- **ImageNet-LT** is a $\rho = 100$ subset of the training set of ImageNet (Deng et al., 2009) ($1,000$ classes), resulting in a dataset size of $115,846$ images.

- **CIFAR10-LT** is a $\rho = 100$ subset of the training set of CIFAR10 (Krizhevsky et al., 2009) (10 classes) with $12,406$ images. Both CIFAR10-LT and ImageNet-LT have high inter-class variability, with their classes having objects of very different scales, textures, and backgrounds.

The Flowers-LT and CIFAR10-LT datasets have $128 \times 128$ and $32 \times 32$ resolution, respectively; all others have a resolution of $64 \times 64$. The varying dataset sizes, domains, numbers of classes, resolutions, feature variabilities, and imbalance ratios allow us to draw important conclusions about the applicability of the methods. We summarize the characteristics of each dataset in Table 1.

Evaluation of the Carnivores, AnimalFaces-LT, Flowers-LT, and iNaturalist2019 datasets is performed on the full dataset, following Shahbazi et al. (2022); Rangwani et al. (2022; 2023); Khorram et al. (2024), as there is no separate validation set large enough to provide information about the quality of the trained models. Since ImageNet-LT and CIFAR10-LT have validation sets with $50,000$ and $10,000$ images, respectively, we use those to compute the metrics described hereinafter.

## 4.3 Evaluation Metrics

In this section, we enumerate the evaluation metrics which we use to assess the performance of the methods. We do not use the Inception Score (IS) (Salimans et al., 2016). The IS was once a very popularly reported metric which has now been shown to not be very informative. Shu et al. (2017) show that, due to its architecture, AC-GAN (Odena et al., 2017) pushes the density of the generator away from the classifier's decision boundary, and likely towards "prettier" samples closer to the mean of each mode, reducing diversity while increasing IS. (Barratt & Sharma, 2018) show that IS "fails to provide useful guidance when comparing models", while Heusel et al. (2017) find that IS is not sensitive to several image corruptions. Finally, Gulrajani et al. (2020) show that IS increases when training data is memorized, which is an undesirable property. We now move to the evaluation metrics that we use:

- **FID:** The Fréchet Inception Distance (Heusel et al., 2017) is the Fréchet distance between the learned and the data distributions, under the assumption that both are normal. It is calculated in the feature space of an InceptionV3 (Szegedy et al., 2016) network trained for classification of ImageNet and is historically the most widely reported metric for generative image models.

Table 2: Results for the Carnivores dataset

| Methods | FID($\downarrow$) | FID$_{\text{CLIP}}$($\downarrow$) | iFID($\downarrow$) | Precision($\uparrow$) | Recall($\uparrow$) | CMMD($\downarrow$) |
|---|---|---|---|---|---|---|
| SG2+ADA | $15.27_{\pm 1.19}$ | $9.76_{\pm 0.08}$ | $48.14_{\pm 2.20}$ | $0.89_{\pm 0.01}$ | $0.09_{\pm 0.02}$ | $36.67_{\pm 0.51}$ |
| Transitional | $\mathbf{11.70}_{\pm 0.55}$ | $\mathbf{7.51}_{\pm 0.72}$ | $\mathbf{42.67}_{\pm 0.75}$ | $0.89_{\pm 0.01}$ | $\mathbf{0.20}_{\pm 0.02}$ | $\mathbf{26.68}_{\pm 1.82}$ |
| LeCam | $48.37_{\pm 30.13}$ | $19.37_{\pm 8.00}$ | $91.47_{\pm 39.09}$ | $0.87_{\pm 0.03}$ | $0.04_{\pm 0.06}$ | $70.96_{\pm 28.80}$ |
| GSR | $16.17_{\pm 0.22}$ | $9.56_{\pm 0.52}$ | $52.33_{\pm 0.36}$ | $0.89_{\pm 0.01}$ | $0.14_{\pm 0.01}$ | $36.88_{\pm 2.49}$ |
| NoisyTwins | $15.57_{\pm 0.12}$ | $9.56_{\pm 0.41}$ | $50.52_{\pm 0.49}$ | $0.89_{\pm 0.01}$ | $0.16_{\pm 0.02}$ | $37.48_{\pm 2.38}$ |
| UTLO | $14.36_{\pm 0.17}$ | $8.79_{\pm 0.14}$ | $48.82_{\pm 1.37}$ | $\mathbf{0.90}_{\pm 0.01}$ | $0.16_{\pm 0.04}$ | $33.72_{\pm 0.82}$ |

- **FID$_{\text{CLIP}}$:** FID is sensitive to the number of samples and that it does not always align with human judgment in non-ImageNet data (Kynkäänniemi et al., 2022). The authors propose using CLIP-ViT (Radford et al., 2021) embeddings instead, and show that they result in a more informative metric. Further work corroborates its usefulness (Rangwani et al., 2023; Jayasumana et al., 2024).

- **iFID** and **iFID$_{\text{CLIP}}$:** The intra-FID (Miyato & Koyama, 2018) measures FID separately for each class and takes their mean. We use both InceptionV3 and CLIP as backbones.

- **CMMD:** Jayasumana et al. (2024) challenge the normality assumption made in FID and compute the (non-parametric) mean maximum discrepancy (MMD) distance in the CLIP embedding space.

- **P&R:** Sajjadi et al. (2018) and Kynkäänniemi et al. (2019) define notions of precision and recall (P&R) for generative models as measures of coverage of the target distribution. Precision estimates how much of the learned distribution overlaps with the data distribution, while recall estimates how much of the training distribution is covered by the learned distribution. Intuitively, precision quantifies the percentage of generated images that could plausibly belong to the training data, while recall quantifies the percentage of training data that could be generated by the GAN.

  In the improved P&R metric (Kynkäänniemi et al., 2019), each manifold is approximated by a union of euclidean balls around each point, whose radius is equal to the distance to that point's $k$-nearest neighbor. Then, precision is the fraction of generated samples which fall within the approximated real manifold, and recall is the fraction of real samples that fall within the approximated learned manifold. These computations are done in the feature space of a pretrained network.

- **D&C:** Density and coverage (Naeem et al., 2020) were proposed as alternatives to precision and recall, respectively. The authors argue that precision and recall are sensitive to outliers and modify the way in which both the real and fake manifolds are computed.

Details about the computation of each metric can be found in the Appendix. For each run, we store model checkpoints every 200 kimg and use the one with the lowest FID for evaluation. For each configuration, we present the mean and standard deviation across three different seeds.

## 5 Results

In this section, we discuss how each method performed with respect to the characteristics of each dataset. Due to table size, we do not report results for iFID$_{\text{CLIP}}$, density, nor coverage in the main paper. Results with the complete set of metrics, as well as generated images for each dataset-method pair can be found in the Appendix.

### 5.1 Transitional-CGAN

The paper that introduced this method did not test it against datasets with significant imbalances and framed its usefulness in the context of limited data (Shahbazi et al., 2022). Indeed, Transitional-CGAN performed exceptionally well in the perfectly balanced and small Carnivores dataset (Table 2), achieving the best values for most metrics, including expressively lower FIDs and higher recall than other methods.

Table 3: Results for the AnimalFaces-LT dataset

| Methods | FID($\downarrow$) | FID$_{\text{CLIP}}$($\downarrow$) | iFID($\downarrow$) | Precision($\uparrow$) | Recall($\uparrow$) | CMMD($\downarrow$) |
|---|---|---|---|---|---|---|
| SG2+ADA | $71.40_{\pm32.62}$ | $14.57_{\pm4.40}$ | $155.50_{\pm32.53}$ | $0.77_{\pm0.06}$ | $0.01_{\pm0.01}$ | $45.46_{\pm18.15}$ |
| Transitional | $24.34_{\pm0.58}$ | $5.88_{\pm0.22}$ | $116.10_{\pm5.58}$ | $0.88_{\pm0.01}$ | $0.15_{\pm0.04}$ | $14.84_{\pm1.08}$ |
| LeCam | $62.13_{\pm9.45}$ | $14.92_{\pm2.13}$ | $153.61_{\pm9.40}$ | $0.78_{\pm0.03}$ | $0.00_{\pm0.00}$ | $50.21_{\pm10.40}$ |
| GSR | $\mathbf{19.77}_{\pm0.40}$ | $\mathbf{4.76}_{\pm0.16}$ | $81.56_{\pm1.13}$ | $0.90_{\pm0.00}$ | $0.12_{\pm0.02}$ | $\mathbf{12.94}_{\pm0.83}$ |
| NoisyTwins | $20.24_{\pm0.95}$ | $5.09_{\pm0.23}$ | $84.29_{\pm3.22}$ | $0.89_{\pm0.01}$ | $\mathbf{0.17}_{\pm0.01}$ | $14.54_{\pm0.99}$ |
| UTLO | $19.92_{\pm0.67}$ | $5.14_{\pm0.41}$ | $\mathbf{80.88}_{\pm3.49}$ | $\mathbf{0.91}_{\pm0.00}$ | $0.10_{\pm0.01}$ | $15.14_{\pm2.13}$ |

Table 4: Results for the Flowers-LT dataset

| Methods | FID($\downarrow$) | FID$_{\text{CLIP}}$($\downarrow$) | iFID($\downarrow$) | Precision($\uparrow$) | Recall($\uparrow$) | CMMD($\downarrow$) |
|---|---|---|---|---|---|---|
| SG2+ADA | $11.62_{\pm3.94}$ | $2.66_{\pm0.85}$ | $99.56_{\pm5.55}$ | $\mathbf{0.85}_{\pm0.02}$ | $0.05_{\pm0.04}$ | $8.01_{\pm2.41}$ |
| Transitional | $19.29_{\pm1.10}$ | $3.24_{\pm0.26}$ | $131.62_{\pm4.16}$ | $0.71_{\pm0.01}$ | $0.09_{\pm0.02}$ | $10.06_{\pm0.65}$ |
| LeCam | $9.71_{\pm0.24}$ | $2.00_{\pm0.01}$ | $96.32_{\pm1.63}$ | $0.84_{\pm0.00}$ | $0.09_{\pm0.02}$ | $6.10_{\pm0.41}$ |
| GSR | $9.26_{\pm1.04}$ | $1.95_{\pm0.07}$ | $96.28_{\pm4.70}$ | $0.82_{\pm0.04}$ | $0.21_{\pm0.04}$ | $6.15_{\pm0.69}$ |
| NoisyTwins | $8.82_{\pm0.19}$ | $2.13_{\pm0.25}$ | $93.23_{\pm1.11}$ | $0.82_{\pm0.01}$ | $0.18_{\pm0.03}$ | $7.29_{\pm1.16}$ |
| UTLO | $\mathbf{7.68}_{\pm0.26}$ | $\mathbf{1.68}_{\pm0.05}$ | $\mathbf{91.51}_{\pm0.99}$ | $0.82_{\pm0.01}$ | $\mathbf{0.27}_{\pm0.02}$ | $\mathbf{5.14}_{\pm0.28}$ |

Table 5: Results for the CIFAR10-LT dataset

| Methods | FID($\downarrow$) | FID$_{\text{CLIP}}$($\downarrow$) | iFID($\downarrow$) | Precision($\uparrow$) | Recall($\uparrow$) | CMMD($\downarrow$) |
|---|---|---|---|---|---|---|
| SG2+ADA | $9.21_{\pm0.15}$ | $2.87_{\pm0.10}$ | $41.09_{\pm0.16}$ | $0.73_{\pm0.00}$ | $0.52_{\pm0.02}$ | $7.63_{\pm0.62}$ |
| Transitional | $12.90_{\pm1.78}$ | $4.04_{\pm0.55}$ | $49.54_{\pm3.63}$ | $0.73_{\pm0.02}$ | $0.45_{\pm0.01}$ | $10.38_{\pm1.77}$ |
| LeCam | $9.27_{\pm0.21}$ | $2.81_{\pm0.11}$ | $\mathbf{41.02}_{\pm0.54}$ | $0.73_{\pm0.01}$ | $0.52_{\pm0.01}$ | $7.49_{\pm0.61}$ |
| GSR | $\mathbf{9.20}_{\pm0.03}$ | $\mathbf{2.69}_{\pm0.08}$ | $41.82_{\pm0.17}$ | $0.72_{\pm0.02}$ | $0.54_{\pm0.03}$ | $\mathbf{6.66}_{\pm0.38}$ |
| NoisyTwins | $9.25_{\pm0.16}$ | $2.90_{\pm0.06}$ | $42.10_{\pm0.35}$ | $0.72_{\pm0.01}$ | $\mathbf{0.55}_{\pm0.00}$ | $7.35_{\pm0.23}$ |
| UTLO | $10.40_{\pm0.64}$ | $3.19_{\pm0.17}$ | $43.20_{\pm1.48}$ | $\mathbf{0.73}_{\pm0.00}$ | $0.50_{\pm0.01}$ | $8.63_{\pm0.82}$ |

Table 6: Results for the iNaturalist2019 dataset

| Methods | FID($\downarrow$) | FID$_{\text{CLIP}}$($\downarrow$) | iFID($\downarrow$) | Precision($\uparrow$) | Recall($\uparrow$) | CMMD($\downarrow$) |
|---|---|---|---|---|---|---|
| SG2+ADA | $2.84_{\pm0.09}$ | $0.84_{\pm0.03}$ | $73.83_{\pm0.51}$ | $0.74_{\pm0.01}$ | $0.65_{\pm0.00}$ | $24.02_{\pm0.33}$ |
| Transitional | $4.02_{\pm0.30}$ | $1.06_{\pm0.13}$ | $92.95_{\pm2.68}$ | $0.73_{\pm0.01}$ | $0.63_{\pm0.01}$ | $25.70_{\pm0.25}$ |
| LeCam | $\mathbf{2.76}_{\pm0.08}$ | $\mathbf{0.84}_{\pm0.04}$ | $\mathbf{72.99}_{\pm0.29}$ | $\mathbf{0.74}_{\pm0.01}$ | $0.65_{\pm0.00}$ | $24.45_{\pm0.30}$ |
| GSR | $3.41_{\pm0.15}$ | $1.00_{\pm0.02}$ | $73.74_{\pm0.19}$ | $0.74_{\pm0.00}$ | $0.64_{\pm0.00}$ | $25.39_{\pm0.45}$ |
| NoisyTwins | $2.99_{\pm0.06}$ | $0.99_{\pm0.07}$ | $76.31_{\pm0.24}$ | $0.73_{\pm0.00}$ | $\mathbf{0.66}_{\pm0.00}$ | $24.93_{\pm0.32}$ |
| UTLO | $3.26_{\pm0.04}$ | $1.03_{\pm0.04}$ | $76.22_{\pm0.43}$ | $0.73_{\pm0.00}$ | $0.65_{\pm0.00}$ | $24.89_{\pm0.25}$ |

Table 7: Results for the ImageNet-LT dataset

| Methods | FID($\downarrow$) | FID$_{\text{CLIP}}$($\downarrow$) | iFID($\downarrow$) | Precision($\uparrow$) | Recall($\uparrow$) | CMMD($\downarrow$) |
|---|---|---|---|---|---|---|
| SG2+ADA | $19.12_{\pm0.47}$ | $6.72_{\pm0.33}$ | $212.56_{\pm1.39}$ | $0.73_{\pm0.02}$ | $0.37_{\pm0.01}$ | $22.46_{\pm1.31}$ |
| Transitional | $19.30_{\pm1.09}$ | $\mathbf{5.46}_{\pm0.32}$ | $230.08_{\pm3.31}$ | $0.71_{\pm0.00}$ | $0.39_{\pm0.02}$ | $\mathbf{16.78}_{\pm1.13}$ |
| LeCam | $\mathbf{18.83}_{\pm0.52}$ | $6.65_{\pm0.05}$ | $\mathbf{210.32}_{\pm1.91}$ | $0.74_{\pm0.01}$ | $0.36_{\pm0.01}$ | $22.01_{\pm0.27}$ |
| GSR | $22.30_{\pm0.43}$ | $7.08_{\pm0.16}$ | $220.94_{\pm2.34}$ | $\mathbf{0.75}_{\pm0.01}$ | $0.31_{\pm0.04}$ | $23.31_{\pm1.04}$ |
| NoisyTwins | $20.39_{\pm0.36}$ | $7.17_{\pm0.39}$ | $218.90_{\pm0.18}$ | $0.73_{\pm0.01}$ | $0.37_{\pm0.01}$ | $24.54_{\pm1.82}$ |
| UTLO | $20.30_{\pm0.45}$ | $7.36_{\pm0.23}$ | $213.52_{\pm0.91}$ | $0.73_{\pm0.01}$ | $\mathbf{0.40}_{\pm0.01}$ | $25.50_{\pm1.16}$ |

In the AnimalFaces-LT dataset (Table 3), which is both limited and long-tailed, Transitional-CGAN still improved significantly over the baseline, despite not achieving FID scores as low as those of GSR and UTLO. This was expected, as these methods were developed specifically to deal with long-tailed class distributions.

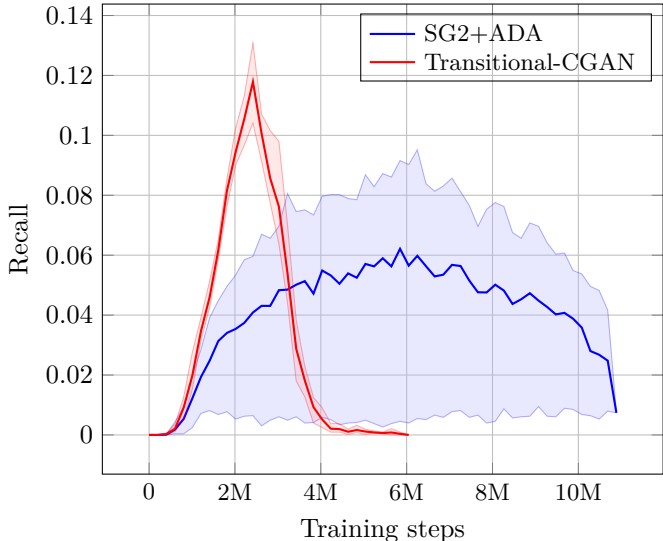

Figure 1: Performance of the SG2+ADA baseline and Transitional-CGAN according to the recall metric. Training steps correspond to the number of real images seen by the discriminator. Note that Transitional-CGAN's transition to conditional training starts at 2 million training steps.

In all other datasets except ImageNet-LT, Transitional-CGAN showed little to no improvement over the baseline.

Flowers-LT has the lowest inter-class variability of all tested datasets. This is a condition where we might think that the feature sharing mechanism of unconditional training would be particularly useful, which clearly was not the case for the trained models (Table 4). However, upon inspection of the recall metric during training, we could see that by the time the transition from unconditional to conditional training started, recall was increasing much faster in Transitional-CGAN than in the baseline (Figure 1). Furthermore, training started to collapse soon after the transition started. We hypothesize that both a delayed transition start and a slower transition to conditional training could have stabilized Transitional-CGAN, yielding results at least as good as the baseline, probably with improvements with regard to variability (i.e., recall).

CIFAR10-LT, on the other hand, has particularly high variability across its 10 classes, as well as $12,406$ images. In this case, the use of Transitional-CGAN was very counterproductive (Table 5), as there was likely little benefit to sharing information between classes.

Both iNaturalist2019 and ImageNet-LT have a large number of samples, so the expectation was that Transitional-CGAN would not be very useful here. Although this was the case for iNaturalist2019 (Table 6), the same cannot be said for ImageNet-LT, particularly the CLIP-based metrics (Table 7).

## 5.2 UTLO

The performance of UTLO (Khorram et al., 2024) went according to expectations: in datasets where image global structure, such as object size and orientation, are similar between classes, UTLO is helpful. By training the first few layers unconditionally, they learn features from all classes. When generating samples for tail classes, these lower resolution features made richer by the head classes induce greater diversity.

UTLO effectively uses those coarse features learned from head classes to add diversity to tail classes.

The Carnivores, AnimalFaces-LT, and Flowers-LT datasets (Tables 2, 3, and 4) generally all have centered objects with similar scale and structure between classes. In these datasets, UTLO can be said to have been a strict improvement over the baseline, achieving some of the best results on several metrics. We highlight that the increases in sample diversity caused by UTLO are reflected in the recall and coverage metrics, which estimate how much of the target distribution is covered by the generator's distribution.

In the other three datasets, which have larger amounts of samples and greater variability in image structure, UTLO generally failed to improve results over the baseline. This was expected as there is less benefit to sharing high-level features among classes in these datasets. For example, the CIFAR-10 classes *airplanes*, *cars*, and *frogs* have very different backgrounds and object scales.

We now refer back to our note in Section 3.2 regarding the possibility of UTLO generating out-of-distribution samples due to the introduction of high-level features (such as pose and orientation) that are unnatural in certain classes. We now discuss UTLO's performance with respect to the precision metric, which estimates how much of the generated data could plausibly have come from the training data.

The only way we believe that UTLO might help prevent the model from generating out-of-distribution examples is indirectly, by stabilizing training. Previous work has shown that conditioning all layers of the generator can lead to instability in spectral norms for underrepresented classes (Rangwani et al., 2022). Furthermore, UTLO's unconditional objective provides stable gradients even when data is scarce. For example, we can see in the AnimalFaces-LT dataset (Table 3) that the baseline models diverged, leading to a worse performance across all metrics. In this dataset, which has both a small number of samples and a long-tailed class distribution, UTLO might have achieved the highest precision by simply stabilizing training the best.

Interestingly, UTLO's precision in Flowers-LT is lower than the baseline's, which actually achieved the highest precision out of all methods (Table 4). However, this is due to a large-scale mode collapse in the baseline models: the model collapsed in several classes, outputting images of very low diversity and very similar to those in the training data (see Appendix). In this scenario, models achieve very high precision simply because they only learn a small part of the data. This can be confirmed from looking at the recall of both models: while the baseline had a very low recall, UTLO achieved the highest.

On the other hand, as previously mentioned, we might expect that leaving the first layers of the generator unconditional might allow it to generate out-of-distribution samples. However, we observed this to happen only to a very small degree: In the datasets where UTLO did not improve performance compared to baseline with respect to the FID-based metrics and CMMD (Tables 6, 5, and 7), it also did not significantly degrade precision and density. Furthermore, visual inspection of generated samples shows that they only rarely exhibit unusual large-scale features for their respective class.

## 5.3 LeCam Regularization

Our experiments paint a very clear picture of the applicability of the LeCam regularizer (Tseng et al., 2021): it can likely be applied in most scenarios, but may require extensive hyperparameter sweeps depending on two key dataset characteristics: degree of inter-class variability and sample size.

In general, we can expect the LeCam regularizer to perform better in scenarios where there is low inter-class variability. This is the case in Flowers-LT and iNaturalist2019 (Tables 4 and 6), where it achieved improvements over the baseline according to almost all metrics. In these scenarios, forcing discriminator predictions to be less extreme may help the generator to learn subtle intra-class features and prevent memorization.

However, the LeCam regularizer achieved comparable results to the baseline in the high-variability CIFAR10-LT and ImageNet-LT datasets (Tables 5 and 7). In these datasets, classes are much more easily separable, so the discriminator's signal is already useful, giving the LeCam regularizer little room to improve training.

Finally, results in the two smaller datasets, Carnivores and AnimalFaces-LT, were significantly worse than baseline (Tables 2 and 3). These datasets have medium to high inter-class variability as well as very limited sizes (2k and $1,756$ samples, respectively). In these cases, where learning signals are already very weak and noisy, the LeCam regularizer may be over-regularizing the model, not allowing it to capture structure. This can be verified by visual inspection of the generated samples (see Appendix).

## 5.4 GSR

The group spectral regularizer (GSR) (Rangwani et al., 2022) behaves very predictably. In the long-tailed and small- and medium-scale AnimalFaces-LT, Flowers-LT, and CIFAR10-LT (Tables 3, 4, 5), it improved

performance in almost all metrics. This was expected, as this method was developed specifically to deal with long-tailed data.

Furthermore, even though AnimalFaces-LT classes share global structure (e.g., scale and position) to some extent, classes differ significantly in terms of intra-class feature variability. For example, dogs and cats have much larger feature variability than tigers. This suggests that, in these scenarios, GSR may be complementary to feature sharing methods such as Transitional-CGAN (Shahbazi et al., 2022) and UTLO (Khorram et al., 2024).

Unsurprisingly, GSR had only a small positive effect in the Carnivores dataset, which is perfectly balanced (Table 2).

Table 8: FIDs of top three head classes and bottom five tail classes on iNaturalist2019

| Methods | Class 000 | Cl. 002 | Cl. 008 | Cl. 881 | Cl. 288 | Cl. 769 | Cl. 972 | Cl. 658 |
|---------|-----------|---------|---------|---------|---------|---------|---------|---------|
| SG2ADA | $\mathbf{7.23}_{\pm 0.81}$ | $\mathbf{3.00}_{\pm 0.22}$ | $\mathbf{5.72}_{\pm 0.48}$ | $16.28_{\pm 2.00}$ | $24.99_{\pm 1.86}$ | $17.56_{\pm 1.94}$ | $15.31_{\pm 1.63}$ | $25.80_{\pm 6.74}$ |
| GSR | $7.42_{\pm 0.34}$ | $3.03_{\pm 0.06}$ | $6.72_{\pm 0.29}$ | $\mathbf{13.75}_{\pm 0.43}$ | $\mathbf{20.20}_{\pm 0.37}$ | $\mathbf{16.33}_{\pm 0.29}$ | $\mathbf{13.76}_{\pm 0.58}$ | $\mathbf{19.00}_{\pm 0.53}$ |

GSR did not improve performance over the baseline in the two datasets with the highest number of classes, iNaturalist2019 and ImageNet-LT ($1,010$ and $1,000$ classes, referred in Tables 6 and 7, respectively). We hypothesize this to be due to excessive penalization of large eigenvalues of the grouped style vectors corresponding to head classes, resulting in reduced feature diversity for these classes. We verify this experimentally by computing class-wise FIDs for head and tail classes on these two datasets: in Table 8, we can see that GSR consistently achieves better FIDs on tail classes but worse FIDs on head classes compared to the baseline. Further evidence supporting our hypothesis of GSR being too aggressive on head classes is that its recall and coverage in these two datasets are always slightly lower than the baseline: due to their frequency in the dataset, head classes likely contribute more to the calculation of the approximate manifold coverage than tail classes do, leading to a decrease in the values of these metrics. Despite weighing its regularization term by the effective number of samples of each class (Eq. 11), GSR's results could likely be improved by hyperparameter search or by more carefully adjusting its strength to each class.

### 5.5 NoisyTwins

NoisyTwins (Rangwani et al., 2023) outperformed the baseline, with respect to most metrics, in all datasets except for iNaturalist2019 and ImageNet-LT (Tables 6 and 7), despite the fact that, in the paper, the authors propose NoisyTwins mainly as a technique to address training conditional GANs on large-scale long-tailed datasets (Rangwani et al., 2023).

However, the reason for this discrepancy in the results is clear: the authors use a batch size of 128 while, for fairness, we use a batch size of 64 across all our experiments. Due to NoisyTwins' contrastive regularization being based on *twin pairs* of style vectors, batch size is effectively reduced in half in terms of class representation. This is particularly important when training models on iNaturalist2019 and ImageNet-LT, since these have a high number of classes ($1,010$ and $1,000$, respectively). In these cases, each batch has relatively low class diversity, so classes that appear less frequently can yield noisy gradients or reduce the effectiveness of NoisyTwins' contrastive regularization.

## 6 Discussion

In this section, we discuss the practical applicability of the studied methods based on the results in Section 5 and discuss several directions for future work.

### 6.1 Practical Considerations

We found that Transitional-CGAN (Shahbazi et al., 2022) reliably improves performance in datasets with limited data, regardless of imbalance. In datasets with more than a few thousand samples, however, it seems

to hinder performance. One exception to this was ImageNet-LT; we leave further investigation of these results for future work.

UTLO (Khorram et al., 2024) works well regardless of dataset size, with its applicability depending only on whether classes share large scale structure. When dealing with these datasets, UTLO can be expected to significantly improve performance.

The LeCam regularizer (Tseng et al., 2021) is useful when inter-class variability is low, i.e., when a large number of classes are very similar. If classes are different enough, penalizing the discriminator makes the model underfit by not allowing it to give strong enough signals to the generator.

The group spectral regularizer (GSR) (Rangwani et al., 2022) generally improved performance over the baseline in all but the large-scale datasets (iNaturalist2019 and ImageNet-LT). Excluding these two datasets, GSR was robust to different levels of imbalance and intra- and inter-class variability. Furthermore, due to its relative insensitivity to its only hyperparameter, we strongly recommend that it be considered as part of baseline models for datasets with characteristics similar to those in our benchmarks.

NoisyTwins (Rangwani et al., 2023) also performed well in all but the large-scale datasets. However, while we believe that GSR's results on these datasets could be improved either by better adjusting its strength by class or by hyperparameter search, we expect NoisyTwins to have improved over the baselines on iNaturalist2019 and ImageNet-LT simply by increasing its batch size to 128, which was used in the original paper. In general, we recommend employing NoisyTwins if it is possible to use a sufficiently large batch size with respect to the number of classes of the dataset.

We note that parameter counts are almost identical across all methods, and that the largest computational overhead with respect to the baseline is an $\approx 6.5\%$ increase in training iteration time. For a more detailed breakdown of the computational budget of each method, we refer the reader to the Appendix.

## 6.2 Future Work

Although each method improved CGAN training in our experimental settings, none fully resolved the underlying challenge, highlighting the need for more generalizable and effective approaches.

Existing metrics for evaluating image generative models, such as FID and iFID, measure global distributional similarity and do not attempt to capture generation quality or diversity within tail classes specifically. This is a limitation of current evaluation practices in long-tailed generative modeling, and the development of metrics that explicitly target tail class performance (for instance by measuring intra-class diversity or quality independently of head classes) represents a meaningful direction for future work.

The expectation regarding Transitional-CGAN was that its knowledge sharing feature might be useful in larger long-tailed datasets with low inter-class variability: if many classes share features to a significant degree, unconditional training might introduce these features to tail classes, increasing their diversity. However, as discussed in Section 5, results on one such dataset, Flowers-LT, did not meet expectations, and we argued that that might be due to the fixed nature of the start and end of the transition from unconditional to conditional training. As such, we suggest investigating an adaptive version of Transitional-CGAN. Ideally, it would take into account whether there is a benefit to prolonging unconditional training (as seemed to be the case with Flowers-LT), whether the transition is done at an appropriate rate, and whether the weights of the conditional and unconditional objectives really both ought to be 1 after the transition has ended.

We were also unable to explain the notable improvements of Transitional-CGAN over the baseline in ImageNet-LT. It may be promising to investigate why this small change to the training process resulted in significant improvements in a large-scale, long-tailed dataset such as ImageNet-LT.

We found that the LeCam regularizer only improved model performance when data had low inter-class variability. However, the idea of penalizing discriminator overconfidence is an interesting one. In particular, there is potential in adapting the LeCam regularizer to scale its penalty in a per-class fashion, striking a balance between allowing the discriminator learn from more diverse classes while not allowing it to get overconfident on classes with few samples or little variety.

Finally, it may be worth exploring how methods in the rich self- and semi-supervised learning literature may be adapted to training CGANs in constrained data settings. Since these methods respectively specialize in learning in the total and very limited presence of supervision, there may be some value to adapting them to the generative paradigm. NoisyTwins is one such example, adapting the Barlow Twins self-supervised learning method to induce class-consistent outputs in CGANs.

### Broader Impact Statement

This paper presents work whose goal is to advance the field of Machine Learning. There are many potential societal consequences of our work, none which we feel must be specifically highlighted here.

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
