# OpenReview forum: "Training Conditional GANs on Limited and Long-Tailed Data: a Survey and Comparative Analysis"
_TMLR — Rejected by TMLR_

### Review · Reviewer_FLXP · 2026-03-15

**Summary Of Contributions:**

This paper presents a survey and comparative empirical analysis of methods for training conditional GANs on limited and long-tailed data. Using StyleGAN2+ADA as the baseline, the authors evaluate five additional techniques: Transitional-CGAN, UTLO, LeCam regularization, GSR, and NoisyTwins. Experiments are conducted on six datasets with different levels of data availability and imbalance, including one balanced dataset and five imbalanced or long-tailed datasets. The evaluation uses six reported metrics: FID, FIDCLIP, iFID, precision, recall, and CMMD. The strengths, weaknesses, and practical applicability of each method are discussed.

I think this paper does a relatively good job of conducting comprehensive and extensive evaluations. To promote fair comparison, the paper adopts StyleGAN2 as a common backbone, uses ADA to build a strong shared baseline, and implements the compared methods within the same overall framework. The study goes beyond reporting aggregate numbers and discusses the practical applicability of each method under different data conditions. For example, the paper finds that Transitional-CGAN is particularly effective in very small and balanced datasets, UTLO is most helpful when classes share similar global structure, and GSR is especially useful in long-tailed settings, although it may hurt head-class performance in datasets with a very large number of classes. This kind of dataset-dependent analysis makes the empirical study more informative for practitioners.

**Audience:**

No

**Audience Explanation:**

The introduction spends a noticeable amount of space defending why the paper studies GANs instead of VAEs or diffusion models, emphasizing that diffusion models are expensive and that GANs remain attractive for practitioners with limited compute. That is a reasonable motivation, but the current framing is a bit too defensive and risks creating the impression that the paper is arguing against the field’s momentum rather than establishing a compelling research problem. This may slightly weaken reader interest rather than strengthen it. The introduction would sound stronger if it provided the context that GANs remain a practically compelling option in several regimes despite the rise of VAEs and diffusion models, and more directly foregrounded the core problem. To better highlight the paper's contribution, the introduction could briefly refer readers to existing surveys on VAEs and diffusion models, and then more clearly articulate the specific gap addressed here: the lack of a unified comparative study of conditional GAN methods under limited and long-tailed data.

**Broader Impact Concerns:**

No special concerns.

**Claims And Evidence:**

No

**Claims Explanation:**

I select no to introduce the following weakness and corresponding requested changes.

Because the paper emphasizes comparative analysis, I think the evaluation part could be further strengthened. Although all methods are built on a shared backbone, additional techniques may introduce additional components or computational overhead. For instance, UTLO adds an unconditional objective and effectively requires extra processing at intermediate resolutions, while NoisyTwins introduces twin augmentations together with an additional correlation-based regularizer. As a result, it is unclear whether some of the observed performance differences are partly influenced by differences in computational budget. The current experiments do not report parameter counts or training time. Including model size and training cost would improve both the fairness of the comparison and the practical value of the paper.

**Requested Changes:**

Could the authors report parameter counts and training cost (e.g., GPU-hours) for each method and discuss whether the improvements come from different computational budgets?

Could the authors revise the introduction based on the above discussions?

---

> ### Author Response · Authors · 2026-03-23
> **Response to Reviewer FLXP**
>
> We start by thanking the reviewer for their comments and acknowledge that the Introduction might have come across as too defensive regarding the field's momentum with respect to diffusion models. We have rewritten the paragraphs discussing diffusion models, VAEs, and GANs after enumerating them in page 2. In our revised Introduction, we:
> - Try to better highlight the capabilities of modern diffusion models, and refer interested readers to two seminal surveys on this topic.
> - Removed a specific direct comparison between GANs and diffusion models, which might have come across as "putting down" recent progress in the latter.
> - Highlight the role of the paper in the literature: "Though GANs are poised as a very appealing choice for generative modeling due to their high fidelity, low inference cost, and greater success with limited data, there is currently no unified study of existing methods for training conditional GANs on limited and long-tailed data."
>
> We also acknowledge that the paper should make computational budget differences between methods clear. To that end, we added a paragraph at the end of Section 6.1 Practical Considerations stating that parameter counts and training iteration times are similar across methods, and refer the reader to the Appendix for a more detailed analysis. In the Appendix, we provide explicit parameter counts, training iteration time differences, and convergence curves for two datasets. We also explain the reasons for the small differences in parameter counts and training iteration time between methods, when applicable.

---

### Review · Reviewer_6Cef · 2026-03-15

**Summary Of Contributions:**

This paper presents itself as a survey on conditional GANs trained on limited data. However, it fails to provide a clear taxonomy or a cohesive view of existing literature encompassing architectures, training frameworks, regularizations, and the effects of data on GANs. Rather than synthesizing a decade of research with the benefit of hindsight, the results read as a disconnected discussion of six methods across several datasets. Furthermore, the experiments do not present fair or comparable results; they lack necessary ablation studies to isolate the effects of specific components like regularization, training frameworks, or other proposed techniques.

While studying GANs in the current era of successful generative methods is an interesting motivation, the evaluation must broadly compare existing literature and discuss its positioning relative to modern alternatives (e.g., diffusion models and their successful techniques). Ultimately, this manuscript lacks the breadth, depth, discussion, and unification expected of a survey paper.

On the positive side, the six methods were evaluated in six datasets using several metrics.  This setup could serve as a base to further do a systematic study that can support the intended survey.

**Audience:**

No

**Audience Explanation:**

While a survey and an cohesive view of methods that did generative modeling is interesting to the TMLR audience, the current form of the paper is not of interest to this audience.  The paper requires extensive work to be a survey that provides the extensive results and discussions claimed in the contributions of the paper.

**Claims And Evidence:**

No

**Claims Explanation:**

The paper claims to provide an extensive literature review, extensive comparative analysis, and a discussion of the practical applicability of the methods.  The paper however, fails to present an extensive literature review, the comparative analysis is also limited to six variants without an explanation of why these variants are representative of the literature and lack a unified view of the field to draw conclusions from.  The comparative analysis is limited to each of the methods and doesn't present a cohesive analysis of the results or the possible methods variations, as well as how these techniques can be applied in contrast to the existing literature and state of the art.

**Requested Changes:**

Comments:

- The paper lacks a discussion of previous GAN surveys and fails to identify how this work fills existing gaps in the literature.
- Wasserstein GANs are omitted; instead, the authors only discuss LeCam regularization.
- The paper fails to reconcile different perspectives into a generalized framework that connects distinct lines of research.
- Results for each method are presented in a vacuum, lacking broader insights or guidance on how to adapt other methods to these setups.
- The experimental section lacks a cohesive discussion that ties together a decade of research on this topic.

Minor Comments:
- Notation needs improvement. Losses are treated as variables and reassigned to themselves (e.g., in Equations 9 and 12). This assignment style is unconventional in mathematical writing and should be revised.


Requested changes:

1. Structure and Taxonomy
- Establish a clear taxonomy: Reorganize the paper to categorize the literature by architectures, training frameworks, regularizations, and data effects. Avoid simply listing the methods sequentially.
- Propose a generalized framework: Synthesize the disparate lines of research into a cohesive framework that logically connects different perspectives and variations.
- Contextualize within existing literature: Add a dedicated subsection reviewing previous GAN surveys and explicitly state the novel contributions and specific gaps this survey addresses.
- Organize the discussion and results on a set of findings that describe the main results in a logical order where the different parts of the unified view of conditional GAN are evaluated.


2. Scope and Modern Relevance
- Position against modern generative models: Broaden the evaluation to include a critical discussion of how conditional GANs (in limited-data regimes) compare to and position themselves against modern alternatives, specifically diffusion models.
- Include foundational architectures: Expand the discussion of regularizations and architectures to include critical developments like Wasserstein GANs, rather than limiting the scope to LeCam regularization, as well as the effect of different backbone architectures for the generator and discriminator including the effect of modern ones like ViTs.

3. Experimental Rigor and Comparability
- Standardize the evaluation: Revise the experimental section to ensure all methods are evaluated under fair, directly comparable conditions across the datasets. Evaluating particular characteristics of the GANs in the different settings would be more benefitial to the reader instead of a single discussion section.
- Conduct ablation studies: Introduce experiments that isolate and evaluate specific components (such as regularization techniques, training frameworks, or other proposed modifications) to understand their individual contributions.
- Extract broader insights: Move beyond reporting individual method results in a vacuum. Provide actionable guidance or theoretical insights on how the setups or techniques discussed can be adapted to other methods.
- Synthesize a decade of research: Expand the final discussion of the experimental results to critically reflect on the historical progression of the field, drawing overarching conclusions and lessons learned.

---

> ### Author Response · Authors · 2026-04-17
> **Response to Reviewer 6Cef [Part 1/2]**
>
> (This comment is the first of a two-part response.)
>
> We would like to start by thanking the reviewer for their time and comments on our paper.
> In this response, we will refer to particular excerpts from the reviewer's comments and address them individually.
>
> "the results read as a disconnected discussion of six methods across several datasets"
> - The purpose of the experimental section is to provide a comparative assessment of the ability of different GAN methods from the literature in the low-data regime. In this context, all methods are evaluated within a common robust experimental backbone, and their individual performance is then discussed. We would like to kindly ask for a clarification regarding what aspects of the discussion feel disconnected, and how the discussion could be framed alternatively.
>
> "The experiments (...) lack necessary ablation studies to isolate the effects of specific components", "Conduct ablation studies: (...) to understand their individual contributions."
> - We would like to re-emphasize that the purpose of the experimental section is to provide a comparative assessment of the performance of different GAN methods from the literature in the low-data regime. An ablation study regarding the impact of different components of these methods in their performance is beyond the scope of the paper; in fact, modifications to the methods (such as removing components) would be tantamount to considering alternative architectures, which are not the focus of our survey. We refer to the original papers, where the GAN methods we discuss in our survey are first proposed, for detailed ablation studies on the role of the different components of the different architectures.
>
> "Rather than synthesizing a decade of research (...)", "The experimental section lacks a cohesive discussion that ties together a decade of research on this topic"
> - While it is perfectly possible that we may have missed important references, to the best of our knowledge the first work to directly identify and address the issues of training conditional GANs on limited data was by Shahbazi et al., in 2022. As such, we would like to kindly request specific pointers to the references that we have missed.
>
> "Wasserstein GANs are omitted; instead, the authors only discuss LeCam regularization."
> - We would like to note that the Wasserstein GANs are not designed to address the issues discussed in our paper. To the best of our understanding, the only connection between the Wasserstein loss used in the training of Wasserstein GANs and our paper is that, when the LeCam regularizer is used with a Wasserstein objective, training amounts to minimizing a LeCam divergence. We also note that the authors of the LeCam regularizer themselves do not use the Wasserstein objective in any of their experiments, and that state-of-the-art backbones (StyleGAN2, BigGAN, R3GAN) have not used the Wasserstein objective in the most recent works. We will add a clarification in this respect to the text in Section 3.3.
>
> "The paper fails to reconcile different perspectives into a generalized framework that connects distinct lines of research", "Results for each method are presented in a vacuum (...)", "Provide actionable guidance or theoretical insights on how the setups or techniques discussed can be adapted to other methods."
> - The paper provides a categorization of different methods besides an individual discussion of their strengths and weaknesses in the low-data regime, backed up by a broad experimental study using a common robust experimental backbone. The use of a common and broad set of experiments allows for a direct comparison between the different methods, thus highlighting their relative merits in the low-data regime. As such, we would like to kindly request the reviewer to provide more specific guidance as to what aspect of our discussion may be unclear, or what aspects the reviewer would like to see further discussed/compared/connected.
>
> "Reorganize the paper to categorize the literature by architectures, training frameworks, regularizations, and data effects", "Include foundational architectures"
> - We note that our survey is not meant to provide a survey of all GAN architectures, but instead to survey representative, state-of-the-art methods to train conditional GANs in limited, long-tailed data. Throughout our discussion, we refer to the excellent taxonomy and benchmark of GANs by Kang et al., 2023. To further reinforce this categorization, we added a reference to the aforementioned work at the start of page 3 in the Introduction. We also clarify, in the Introduction, that "our objective in this work is to analyse state-of-the-art methods to train conditional GANs in limited and long-tailed data".

---

> > ### Author Response · Authors · 2026-04-17
> > **Response to Reviewer 6Cef [Part 2/2]**
> >
> > (This comment is the second of a two-part response.)
> >
> > "Organize the discussion and results on a set of findings that describe the main results in a logical order where the different parts of the unified view of conditional GAN are evaluated."
> > - We organize our discussion of results in two different ways: in Section 5, we discuss results separately for each method; in Section 6.1, we follow up with a discussion of the practical applicability of the different methods. In our perspective, this organization provides, at the same time, a detailed discussion of the different methods and a broad view on their relative methods. However, we would be thankful for specific suggestions of alternative organizations that could help improve the clarity and/or depth of the discussion.
> >
> > "Revise the experimental section to ensure all methods are evaluated under fair, directly comparable conditions across the datasets."
> > - As mentioned above, all methods are evaluated within a common robust experimental backbone, which enables a direct comparison for the performance of the different methods.
> >
> > "Losses are treated as variables and reassigned to themselves (...)"
> > - We thank the reviewer for this observation. We acknowledge that the assignment style adopted in the paper is unconventional, and have rewritten equations 9, 12, and 15 so that they now use a \leftarrow to make the assignment more explicit.

---

> > > ### Comment · Reviewer_6Cef · 2026-05-07
> > >
> > > I thank the authors for their replies.
> > >
> > > While I understand the authors point of view, I still maintain that the paper requires significant changes to be of interest to the broader community given its narrower scope.
> > >
> > > I was waiting for the other reviews to appear before commenting to get a grasp of what others have to say.  Yet, this hasn't happened, I don't want to leave the authors without a reply.
> > >
> > > I will revisit the review and the comments when the other reviews appear.

---

> > > > ### Comment · Reviewer_6Cef · 2026-05-25
> > > >
> > > > After reading the other reviews and the changes provided by the authors, I maintain my concerns about the paper needing significant changes not presented in this revised version.  Moreover, I agree with the other reviews that the evidence and claims are not convincing.

---

### Review · Reviewer_L2LP · 2026-05-14

**Summary Of Contributions:**

This paper presents a survey and comparative analysis of methods designed to improve the performance of conditional StyleGANs when trained on limited and long-tailed datasets.

Conditional GANs (consisting of a generator and an adversary trained to distinguish between real and generated samples) can suffer from discriminator overfitting when only a small number of training examples are available for a class, causing the generator to reproduce memorized samples instead of learning a generalized data distribution. This problem becomes particularly severe in long-tailed datasets, where minority classes contain substantially fewer samples than majority classes, increasing the risk of “mode collapse” and poor sample diversity.

Experimentally, the paper compares five methods against a baseline approach based on data augmentation. Two of the evaluated methods attempt to leverage features shared across categories, while the remaining approaches introduce additional regularization terms into the training objective. The methods are evaluated on datasets with varying sizes and different degrees of class imbalance. Subsequently, the authors derive recommendations regarding the suitability of the different approaches under specific dataset characteristics.

---

**Strengths:**

**Survey:** The paper provides a clear and well-written survey of existing approaches, including their conceptual motivation and historical development.

**Selection of experiments**: The paper selects representative and competitive methods for comparison and evaluates them across datasets with varying size and class imbalance characteristics. This enables practical recommendations regarding the suitability of different approaches under different data characteristics.

**Code base**: The authors propose to publish their code base containing all evaluated methods, which supports reproducibility.

---

**Weaknesses:**

**Narrow problem formulation:**
The paper never explicitly defines what is meant by “long-tailed” data. The considered setting appears to focus specifically on datasets with either limited overall sample size or strong class imbalance. In machine learning literature, “long-tailed” indeed  commonly refers to class-frequency imbalance with rare categories, but in statistics the term more generally describes distributions with unusually frequent extreme events or outliers. The paper exclusively considers the former interpretation. As a result, potentially relevant settings involving non-typical or outlier samples within majority classes are not addressed.

**Choice of evaluation metrics and interpretation of results:**
The paper correctly identifies diversity within minority classes as a central challenge of long-tailed generative modeling. However, the experimental evaluation does not include a metric specifically designed to assess diversity or coverage within rare categories. Instead, substantial emphasis is placed on the FID metric, which evaluates similarity between feature distributions using a Gaussian approximation. This raises concerns regarding its suitability as the primary metric in a setting explicitly characterized by highly imbalanced and potentially non-Gaussian data distributions.

At the same time, recall values remain consistently low, in some cases extremely low. Since recall measures the coverage of the target distribution, it may be particularly important in the context of severe class imbalance and rare-category generation. The reported results therefore suggest that none of the evaluated methods fully addresses the underlying challenge. This raises questions regarding either the effectiveness of the methods themselves or the chosen training and hyperparameter settings. This limitation is not discussed, despite potentially being one of the central findings of the experimental evaluation.

**Additional Comments:**

I am not entirely familiar with the broader research landscape. Hence, I cannot conclusively evaluate the completeness of the survey or the novelty of the comparative analysis. However, a brief review of related literature supports the authors’ claim that the paper addresses a comparative gap in the literature.

**Audience:**

Yes

**Audience Explanation:**

Limited and long-tailed (imbalanced) datasets are common in practical applications. An overview of existing approaches for improving GAN training under such conditions, together with guidance on method selection, is therefore highly relevant for machine learning practitioners. At the same time, a comprehensive survey of this kind is also valuable for the research community as a foundation for developing improved methods.

**Broader Impact Concerns:**

No concerns

**Claims And Evidence:**

No

**Claims Explanation:**

Most statements are well-supported by experimental evidence. However, as discussed in the weaknesses section above, the evaluation does not directly measure the core claim of improved diversity in rare classes under long-tailed conditions. While the paper examines several methods in such settings, the ability of these methods to generate diverse samples for minority categories is not explicitly assessed. None of the reported metrics directly capture this aspect. Instead, the analysis relies heavily on the FID metric, which may have limited suitability for capturing class-conditional diversity in highly imbalanced settings. The recall metric is closer to this objective but remains consistently low across methods.

Consequently, it is not fully clear from the presented results to what extent claims such as “feature sharing methods also demonstrate improved generative capabilities for the tail of the data distribution” are empirically substantiated.

**Requested Changes:**

**Major:**

- Include an explicit metric for diversity and coverage of rare categories. In addition, discuss the consistently low recall values observed across all methods and what this implies for minority-class generation performance.

- For each dataset, clearly state in a structured form (e.g., table) which aspect of the evaluation it targets (e.g., class balance, degree of imbalance, data scarcity, inter-class similarity). This information is currently distributed across the results section and should be made more explicit for comparability. Similarly, all table captions should indicate the experimental setting being evaluated.

- For the P&R and D&C metrics, provide a clear definition and explanation in the main text rather than relying on the appendix. It should be explicit how these metrics measure distributional overlap, as this is important for interpretability of the results.

- Clarify the stopping criteria used in the experiments to ensure consistency between Figure 1 and the results reported in Table 3 (according to the text, the dataset of Table 3 belongs to Figure 1, which may also be indicated in the caption).

- Describe in the main text how multiple runs differ in the calculation of mean and standard deviation.

---

**Minor:**
- In Section 2.1, the expression for recovering the original GAN loss appears to be incorrect. The correct formulation should be (according to my calculations) $f(x)= - \\log⁡(1+e^{-x})=\\log⁡(\\frac{1}{1+e^{-x}})$

- For the Barlow Twins method, clarify the meaning of “twin” augmentations and whether these refer to arbitrary paired augmentations of the same input or a specific augmentation strategy.

- CMMD values are reported to be approximately three times the values for FID_CLIP. Consider whether CMMD should be excluded from the table rather than moving density and coverage to the appendix

- The LeCam regularizer is missing from the list of models in Section 4.1

-The experimental comparison includes methods primarily from 2022–2024. It may be worthwhile to verify whether more recent approaches exist that should be included for a more up-to-date comparison.

-The imbalance ratio is defined as the ratio between the largest and smallest class sizes. Please clarify how intermediate class sizes are chosen.

---

> ### Author Response · Authors · 2026-05-20
> **Response to Reviewer L2LP [Part 1/2]**
>
> (This comment is the first of a two-part response.)
>
> We thank the reviewer for investing their time on reading our paper and for their comments. We will refer to particular excerpts from the reviewer's comments and address them individually.
>
> "The paper never explicitly defines what is meant by long-tailed data (...)"
> - Though we do give a more specific definition of what we mean by long-tailed data in Section 2.4., we agree that this is an important definition which should be established earlier in the paper. As such, we now clearly define what is meant by long-tailed data in the Introduction.
>
> "the experimental evaluation does not include a metric specifically designed to address diversity or coverage with rare categories"
> - While we do not present any metric computed specifically for rare categories, we note that the intra-class FID (iFID) is the unweighted average of per-class FIDs. As such, while FID may be more influenced by classes with a larger number of samples, in long-tailed datasets iFID is actually dominated by tail classes in the sense that the majority of classes have few samples. Since the coverage of each class is reflected in its FID through the shape of the Gaussian, we argue that iFID is an adequate proxy for rare class coverage. Finally, we add that though we considered computing recall for only a subset of the classes, this estimate is too noisy since the $k$-NN radius is very large in sparse regions. Indeed, we scanned the literature and could not find any paper where such a metric was reported, presumably for this reason.
>
> "substantial emphasis is placed on the FID metric, which evaluates similarity between feature distributions using a Gaussian approximation"
> - We report FID as a primary metric due to it being the standard for image models, including those trained on limited and long-tailed datasets (please see, e.g., UTLO by Khorram et al., 2024). Nevertheless, we agree with the reviewer's reasoning, which is why we also report CMMD (Jayasumana et al., 2024), which is non-parametric. We find that CMMD is strongly correlated to FID.
>
> "The reported results therefore suggest that none of the evaluated methods fully addresses the underlying challenge."
> - We agree with the reviewer that none of the evaluated methods fully address the underlying challenge and have, in a revision, added this context in Section 6.2. Future Work.
>
> "This raises questions regarding either the effectiveness of the methods themselves or the chosen training and hyperparameter settings. This limitation is not discussed, despite potentially being one of the central findings of the experimental evaluation.", "In addition, discuss the consistently low recall values observed (...)"
> - Regarding hyperparameter settings, as noted in the Supplementary Material, we used the configurations reported by Rangwani et al., 2023 and performed a grid search with datasets that were not in their paper. We found that the results were not very sensitive to these choices. Regarding the low recall values observed, we note that these results are somewhat expected. For example, StyleGAN2+ADA on CIFAR10 (the full, balanced dataset) achieves a recall of only 0.69 (Kang et al., 2023), and 0.52 on CIFAR10-LT due to the challenge of the data imbalance.
>
> "For each dataset, clearly state in a structured form (e.g., table) which aspect of the evaluation it targets (...)"
> - We agree, and have added a table containing relevant information about each dataset in order to make this aspect clearer. We would like to ask the reviewer whether they maintain that each table caption should also contain this information, now that it can be readily consulted using this table.
>
> "For the P&R and D&C metrics, provide a clear definition and explanation in the main text rather than relying on the appendix."
> - We agree, and have added the definition and explanation of P&R to the main paper. However, since we do not report D&C in the main paper, we propose to leave their definition in the Appendix.
>
> "Clarify the stopping criteria used in the experiments (...)"
> - We have added this explanation at the end of Section 4.3. Evaluation Metrics.
>
> "Describe in the main text how multiple runs differ in the calculation of mean and standard deviation."
> - It is not clear to us what this comment is suggesting, and we would like to kindly ask the reviewer to clarify.
>
> "In Section 2.1, the expression for recovering the original GAN loss appears to be incorrect."
> - We confirm that your suggestion is correct, and have made the according change.
>
> "For the Barlow Twins method, clarify the meaning of “twin” augmentations (...)"
> - We have clarified, when explaining NoisyTwins, that the "twin" augmentations refer to two vectors resulting from adding two randomly sampled noise vectors to a common one-hot class vector.

---

> > ### Author Response · Authors · 2026-05-20
> > **Response to Reviewer L2LP [Part 2/2]**
> >
> > (This comment is the second of a two-part response.)
> >
> > "CMMD values are reported to be approximately three times the values for FID_CLIP. Consider whether CMMD should be excluded from the table rather than moving density and coverage to the appendix"
> > - Since we could not fit all the metrics nicely in the main paper, we chose to leave density and coverage to the appendix since, like precision and recall, they measure fidelity to and coverage of the target distribution. On the other hand, we thought that CMMD might be an interesting metric to include in the main paper since it challenges the normality assumption of FID, the most commonly reported metric for image models, even if it is highly correlated to FID and FID_CLIP.
> >
> > "The LeCam regularizer is missing from the list of models in Section 4.1"
> > - We have added the method to the list of models.
> >
> > "The experimental comparison includes methods primarily from 2022–2024 (...)"
> > - To the best of our knowledge, there are no relevant methods more recent than UTLO (Khorram et al., 2024).
> >
> > "(...) Please clarify how intermediate class sizes are chosen."
> > - We have added this explanation to Section 4.2. Datasets when introducing the concept of imbalance ratio.

---

> > > ### Comment · Reviewer_L2LP · 2026-05-21
> > >
> > > I thank the authors for their revisions to the manuscript.
> > >
> > > I appreciate the additional clarification regarding what the authors consider “long-tailed” data, described as “datasets in which a significant fraction of samples belong to a few head classes, while a vast majority of tail classes have a relatively much smaller amount of samples.” Nevertheless, I would have appreciated a more precise criterion distinguishing long-tailed datasets from merely imbalanced ones.
> > >
> > > Independently of my evaluation of the paper, I would like to make the following observation regarding terminology: Section 4.2 explains that intermediate classes are constructed to follow an exponential distribution of samples across classes. In statistics, long-tailed distributions are commonly considered a subclass of heavy-tailed distributions, characterized by slowly decaying tails and a comparatively high probability of extreme events relative to Gaussian distributions. The terminology used in the machine learning literature therefore differs from the statistical interpretation of long-tailed distributions, making a precise definition particularly important.
> > >
> > > ---
> > >
> > > My main concern remains the evaluation based primarily on FID/iFID metrics. Given the narrow scope of the survey, which specifically targets long-tailed datasets, I would expect a more detailed evaluation focused on the central challenge of this setting: generation quality and diversity within tail classes. However, such an analysis is largely missing. The considered metrics either evaluate global distributional similarity or appear to indicate limited performance on tail classes. From my understanding, the comments of Reviewer 6Cef raise a related concern: if the evaluation relies primarily on metrics commonly used for general GAN evaluation, one might also expect comparisons against a broader range of GAN architectures and methods.
> > >
> > > ---
> > > Regarding my suggestion to “Describe in the main text how multiple runs differ in the calculation of mean and standard deviation”: Tables 3–7 report mean values together with standard deviations, but I could not find what these quantities are aggregated over. If they are computed across repeated experiments, it would be helpful to clarify which aspects differ between runs.
> > >
> > > I further appreciate the addition of Table 1. As an additional suggestion, it may be helpful to indicate, either within the table or in the captions, the specific characteristics each dataset is intended to evaluate (e.g., degree of imbalance, limited-data regime, low inter-class variability, or semantic similarity between classes). This would make the role of the individual datasets within the comparative evaluation easier to interpret.

---

> > > > ### Author Response · Authors · 2026-05-26
> > > > **Response to Reviewer L2LP**
> > > >
> > > > We again thank the reviewer for their continued engagement and reading of the revised manuscript, and hope to address their remaining concerns in this revision.
> > > >
> > > > We have added to the introduction that "long-tailed" learning, as used in this article, is an instance of imbalanced learning. However, no universally established criterion distinguishes long-tailed from merely imbalanced datasets: seminal works in deep long-tailed learning (Zhang et al., 2023) do not attempt to define such a cutoff, and the boundary is informal in the literature.
> > > >
> > > > We also thank the reviewer for the observation regarding statistical terminology. We have added a remark at the end of the first paragraph of section 2.4. The Long-Tailed Data Problem acknowledging that the ML usage of "long-tailed" differs from the statistical notion of heavy-tailed distributions, and that our usage follows the established ML convention. We hope this contextualizes the terminology for readers coming from a statistics background.
> > > >
> > > > Given these clarifications, we would like to ask the reviewer whether the definition in the current revision adequately captures the scope of the term as used in this survey.
> > > >
> > > > ---
> > > >
> > > > We agree that FID and iFID are global distributional metrics, but would like to emphasize that in long-tailed datasets, since iFID is computed per class and aggregated, the greater number of tail classes means they collectively carry more weight in the final score. Nevertheless, the fact that no existing metric attempts to capture quality and diversity in tail classes specifically is a limitation of our evaluation and warrants future work. As such, we have added this note as a second paragraph to section 6.2. Future Work, and would like to ask the reviewer whether this framing captures their concern regarding the limitations of existing metrics.
> > > >
> > > > Our choice of evaluation metrics was guided by the fact that they are the only metrics consistently reported across the surveyed methods, making cross-method comparison possible.
> > > >
> > > > Regarding the broader concern about GAN comparisons: the scope of this survey is intentionally restricted to methods specifically designed for long-tailed generation. Including general GAN architectures would conflate two different problem settings and obscure the contribution of the tail-specific designs we analyze. We have clarified this decision more explicitly in the first paragraph of section 4. Methodology.
> > > >
> > > > Furthermore, we would like to draw the reviewer's attention to the supplementary material, which includes grids of generated images for all classes, including tail classes, providing a visual, per-class analysis of generation quality which we hope addresses the reviewer's concern regarding per-class visual analysis.
> > > >
> > > > ---
> > > >
> > > > We thank the reviewer for their clarification and note that we have added, in the last paragraph of Section 4.3. Evaluation Metrics, that the statistics presented in the tables are computed across several seeds.
> > > >
> > > > ---
> > > >
> > > > We thank the reviewer for their suggestion, and have added a column to Table 1 describing the characteristics of each dataset.

---

### Comment · Action_Editor_ZVVw · 2026-05-14
**Author discussion**

Dear Authors and Reviewers,

We have all three reviews in. Therefore, we can start a 2-week author discussion period. Please use this time to engage in a discussion to solve potential issues and improve clarity of statements and questions.

After that, the reviewers are expected to provide their official recommendations. Once we have that, I will provide a decision recommendation. The timeline is the following:
- Now - the end of week 2 from now: the author discussion
- The beginning of week 3 - the end of week 4 from now: reviewers provide official recommendations

We aim to give the decision recommendation in max. 4 weeks from now, but ideally earlier (~ week 3 from now).

Best,
AE

---

### Decision · Action_Editor_ZVVw · 2026-06-18

**Recommendation:** Reject

**Additional Comments:**

N/A

**Audience:**

Yes

**Audience Explanation:**

Yes, but primarily a subset of the TMLR audience. The topic of conditional GAN training under class imbalance is relevant to researchers interested in generative modeling, long-tailed learning, and data scarcity. The paper also attempts to synthesize existing approaches in this area, which could provide value to readers entering the field. However, reviewers generally viewed the scope as relatively narrow and noted that the survey is not sufficiently comprehensive to serve as a definitive reference. Combined with the lack of direct evaluation of tail-class performance and the absence of comparisons to alternative generative modeling approaches, the work's broader impact and appeal are constrained. Consequently, while some specialists may find the findings informative, the expected level of interest across the wider TMLR readership is limited.

**Claims And Evidence:**

No

**Claims Explanation:**

While the paper addresses an interesting and relevant problem setting, namely, training conditional GANs under severe class imbalance, the reviewers identified substantial weaknesses in the evidence supporting its claims. A central concern is that the evaluation does not directly assess the core challenge of the setting, namely the generation of diverse and representative samples for minority classes. The experiments rely primarily on standard GAN evaluation metrics and do not include dedicated measures of diversity or coverage within tail classes. As a result, reviewers found that claims regarding improved generative performance on the tail of the distribution are not sufficiently substantiated. Additional concerns include the lack of comparisons with alternative generative models such as VAEs and diffusion models despite claims suggesting the superiority of GAN-based approaches in long-tailed regimes. Furthermore, the paper's stated goal of providing an extensive survey and comparative analysis is not fully realized, as several reviewers found the literature coverage and empirical comparisons too limited to support the breadth of the conclusions drawn.